# Deductive Verification of Chain-of-Thought Reasoning

**Zhan Ling**[1]* **Yunhao Fang**[1]* **Xuanlin Li**[1] **Zhiao Huang**[1] **Mingu Lee**[2]
**Roland Memisevic**[2] **Hao Su**[1]
[1]UC San Diego, [2]Qualcomm AI Research[†]

## Abstract

Large Language Models (LLMs) significantly benefit from Chain-of-Thought (CoT) prompting in performing various reasoning tasks. While CoT allows models to produce more comprehensive reasoning processes, its emphasis on intermediate reasoning steps can inadvertently introduce hallucinations and accumulated errors, thereby limiting models' ability to solve complex reasoning tasks. Inspired by how humans engage in careful and meticulous deductive logical reasoning processes to solve tasks, we seek to enable language models to perform *explicit and rigorous deductive reasoning*, and also ensure the *trustworthiness* of their reasoning process through self-verification. However, directly verifying the validity of an entire deductive reasoning process is challenging, even with advanced models like ChatGPT. In light of this, we propose to decompose a reasoning verification process into a series of step-by-step subprocesses, each only receiving their necessary context and premises. To facilitate this procedure, we propose **Natural Program**, a *natural language-based* deductive reasoning format. Our approach enables models to generate precise reasoning steps where subsequent steps are more rigorously grounded on prior steps. It also empowers language models to carry out reasoning self-verification in a *step-by-step* manner. By integrating this verification process into each deductive reasoning stage, we significantly enhance the rigor and trustfulness of generated reasoning steps. Along this process, we also improve the answer correctness on complex reasoning tasks. Code will be released at `https://github.com/lz1oceani/verify_cot`.

## 1 Introduction

The transformative power of large language models, enhanced by Chain-of-Thought (CoT) prompting [50, 21, 59, 42], has significantly reshaped the landscape of information processing [14, 26, 49, 56, 13, 55, 23, 29], fostering enhanced abilities across a myriad of disciplines and sectors. While CoT allows models to produce more comprehensive reasoning processes, its emphasis on intermediate reasoning steps can inadvertently introduce hallucinations [4, 30, 16, 20] and accumulated errors [4, 51, 1], thereby limiting models' ability to produce cogent reasoning processes.

In fact, the pursuit of reliable reasoning is not a contemporary novelty; indeed, it is an intellectual endeavor that traces its roots back to the time of Aristotle's ancient Greece. Motivated by the desire to establish a rigorous reasoning process, in his "Organon," Aristotle introduced principles of *logic*, in particular, syllogism, a form of logical argument that applies deductive reasoning to arrive at a conclusion based on two or more propositions assumed to be true. In disciplines that rigorous reasoning is critical, such as judical reasoning and mathematical problem solving, documents must be written in a formal language with a logical structure to ensure the validity of the reasoning process.

---

*Equal contribution

†Qualcomm AI Research is an initiative of Qualcomm Technologies, Inc
All datasets and models were solely downloaded and evaluated by the University of California San Diego.

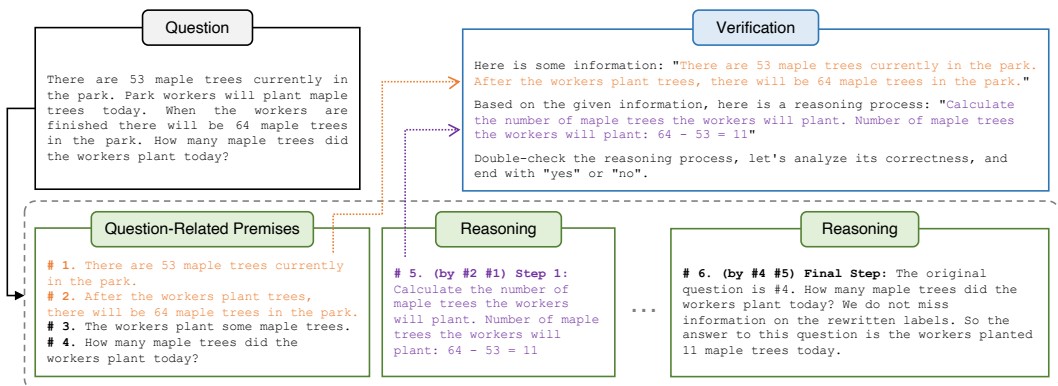

Figure 1: An overview of our proposed deductive reasoning and verification process. In response to an input question, LLMs generate deductive reasoning chains using the *Natural Program* format (bottom 3 boxes), a natural language-based deductive reasoning approach. The Natural Program format allows individual reasoning steps (an example in purple) and their corresponding minimal set of premises (an example in orange) to be easily extracted. This streamlined extraction process facilitates the step-by-step decomposition and verification of deductive reasoning (top-right box).

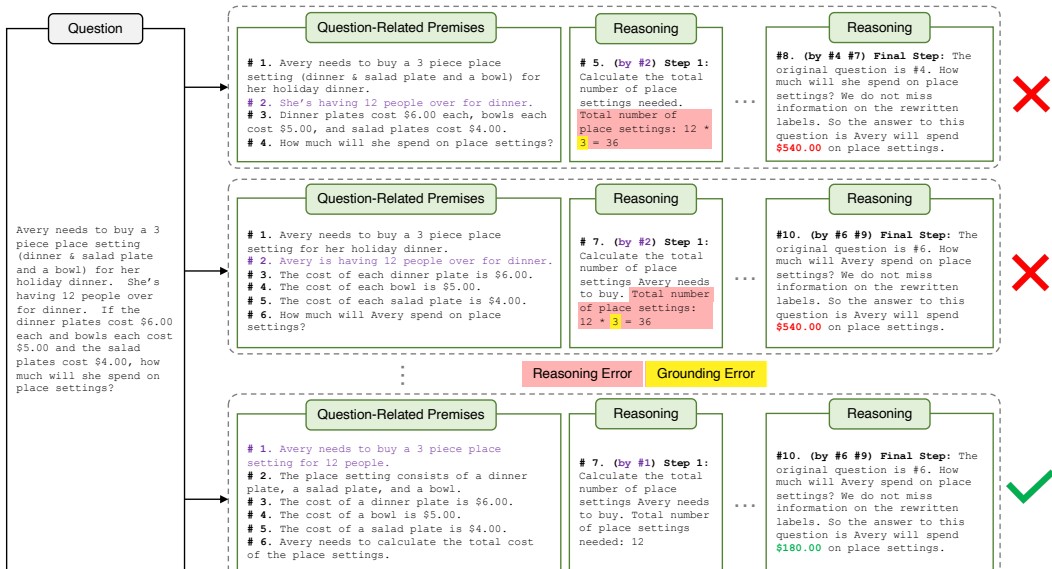

Figure 2: Through our Natural Program-based deductive reasoning verification approach, we identify and eliminate reasoning chains that contain errors in reasoning and grounding (we define grounding error as utilizing information that is not present in cited premises). By alleviating such errors, we significantly enhance the rigor, trustworthiness, and interpretability of the generated reasoning outputs.

We yearn for this sequence of reliable knowledge when answering questions. Our goal is to develop language models that can propose potential solutions through reasoning in logical structures. Simultaneously, we aim to establish a verifier capable of accurately assessing the validity of these reasoning processes. Despite recent significant explorations in the field, such as [48]'s emphasis on self-consistency and [27, 5]'s innovative use of codes to represent the reasoning process, these approaches still exhibit considerable limitations. For example, consistency and reliability are not inherently correlated; as for program codes, they are not powerful enough to represent many kinds of reasoning process, e.g., in the presence of quantifiers ("for all", "if there exists") or nuances of natural language (moral reasoning, "likely", . . . ).

We propose leveraging the power of natural language to achieve the deductive reasoning emphasized in ancient Greek logic, introducing a *"natural program"*. This involves retaining natural language for its inherent power and avoiding the need for extensive retraining with large data sets. A natural program

represents a rigorous reasoning sequence, akin to a computer program. We expect implementations of the idea to have two properties: 1) that natural programs are generated with minimal effort from an existing language model capable of CoT reasoning, preferably through in-context learning; 2) that the natural program can be easily verified for reliability in the reasoning process.

Through a step-by-step investigation, we discovered that large language models have the potential to meet our expectation. Naïve CoT prompts like "Let us think step by step." has many flaws, and entrusting the entire verification process to a large model like ChatGPT can still lead to significant error rates. However, we found that, if the reasoning process is very short, and only based on necessary premises and contexts, the verification of existing large language models is already quite reliable. Therefore, our approach is to design prompts that induce CoT processes comprised of rigorous premises/conditions and conclusions with statement labels, and verification can be done by gradually isolating very few statements within the long thought chain. Experimentally, we found that most reasoning that passed the verification was rigorous, and many that did not pass had elements of imprecision in the reasoning process, even if they occasionally arrived at correct answers.

It is worth emphasizing that, we are not looking for a method to just maximize the correctness rate of final answers; instead, we aspire to generate a cogent reasoning process, which is more aligned with the spirit of judical reasoning. When combined with sampling-based methods, our method can identify low-probability but rigorous reasoning processes. When repeated sampling fails to yield a rigorous reasoning process, we can output "unknown" to prevent hallucinations that mislead users.

We demonstrate the efficacy of our natural program-based verification approach across a range of arithmetic and common sense datasets on publicly-available models like OpenAI's GPT-3.5-turbo. Our key contributions are as follows:

1. We propose a novel framework for rigorous deductive reasoning by introducing a "**Natural Program**" format (Fig. 1), which is suitable for verification and can be generated by just in-context learning;

2. We show that reliable self-verification of long deductive reasoning processes written in our Natural Program format can be achieved through step-by-step subprocesses that only cover necessary context and premises;

3. Experimentally, we demonstrate the superiority of our framework in improving the rigor, trustworthiness, and interpretability of LLM-generated reasoning steps and answers (Fig. 2).

## 2 Related work

**Reasoning with large language models.** Recent large language models (LLMs) [3, 8, 57, 47, 38, 18, 9, 37] have shown incredible ability in solving complex reasoning tasks. Instead of letting LLMs directly generate final answers as output, prior work have shown that by encouraging step-by-step reasoning through proper prompting, such as Chain-of-Thought (CoT) prompting [50] and many others [21, 59, 58, 44, 48, 60, 25, 54], LLMs exhibit significantly better performance across diverse reasoning tasks. To further improve the step-by-step reasoning process, some recent studies have investigated leveraging external solvers such as program interpreters [39, 5, 27], training and calling external reasoning modules [11], or performing explicit search to generate deductive steps [2, 46]. Parallel to these works, we do not rely on external modules and algorithms, and we directly leverage the in-context learning ability of LLMs to generate more precise and rigorous deductive reasonings.

**Large language models as verifiers.** Using language models to evaluate model generations has been a long standing idea [22, 36, 40, 4]. As LLMs exhibit impressive capabilities across diverse tasks, it becomes a natural idea to use LLMs as evaluation and verification tools. For example, [10, 11, 33] finetune LLMs to verify solutions and intermediate steps. LLMs aligned with RLHF [32, 31, 48] have also been employed to compare different model generations. In addition, recent works like [43, 52, 28, 6] leverage prompt designs to allow LLMs to self-verify, self-refine, and self-debug without the need for finetuning. However, these works do not focus on the rigor and trustworthiness of the deductive reasoning processes at every reasoning step. In this work, we propose a natural language-based deductive reasoning format that allows LLMs to self-verify *every* intermediate step of a deductive reasoning process, thereby improving the rigor and trustfulness of reasoning.

**Question**: Jerry is rolling a six-sided die. How much more likely is it (expressed as a percentage) that he rolls a number greater than $3 than that he rolls two even numbers in a row?
**Ground Truth**: There are 3 numbers greater than 3 on the dice, so the chances of rolling one of them are *3 / 6 = 50%*. The chance of rolling one even number is 50%, so the chance of rolling two in a row is *50% \* 50% = 25%*. The difference between these two probabilities is *50% - 25% = 25%*. #### 25

**Output**: There are two possible outcomes for rolling a number greater than 3 out of 6 total outcomes, so the probability is *2 / 6 or 1 / 3*. For rolling two even numbers in a row, there are three possible outcomes (2-4, 4-6, 6-2) out of 36 total outcomes, so the probability is *3 / 36 or 1 / 12*. To find the difference in probability, we subtract the probability of rolling two even numbers in a row from the probability of rolling a number greater than 3: *1 / 3 - 1 / 12 = 1 / 4*. This means that it is 25% more likely to roll a number greater than 3 than to roll two even numbers in a row.

Table 1: An example question from GSM8K with a generated CoT reasoning path with GPT-3.5 (turbo), where the output provides a wrong reasoning chain with the correct answer.

Additionally, while some recent works [12, 53, 15, 34] have proposed methods to verify individual steps in a reasoning process, our approach distinguishes from these works in the following perspectives: **(1)** Our approach leverages in-context learning to achieve reasoning verification, without the need for language model finetuning. **(2)** Our Natural Program-based LLM verification approach not only identifies invalid reasoning steps, but also provides explicit explanations for why they are invalid, detailing the specific reasoning errors involved. **(3)** Our Natural Program-based reasoning and verification approach is compatible with in-context abstract reasoning tasks where reasoning steps do not possess proof-like entailment structures. For example, our approach is compatible with the Last Letters task, where the LLM is instructed to output the concatenation of the last letters of all words in a sequence as the final answer. **(4)** Our Natural Program approach allows the use of commonsense knowledge not explicitly listed in premises. For example, consider this problem: "Marin eats 4 apples a day. How many apples does he eat in November?" Even though "November has 30 days" is not explicitly listed in the premises, Natural Program permits the use of such common knowledge within a reasoning step. Our in-context verification process is also capable of handling these implicit premises (e.g., if LLM outputs "November has 29 days" in a reasoning step, it will be marked as invalid).

## 3   Motivation and Problem Formulation

A reasoning-based question-answering (QA) task can be defined as a tuple $(Q, C, O, A)$ [35], where $Q$ is the target question; $C$ is the context of a question, such as the necessary background for answering a question; $O = (o_1, o_2, \cdots, c_k)$ are optional answer choices if $Q$ is a K-way multiple choice problem; and $A$ is the ground-truth answer. Given $Q$ and $C$ as inputs, large language models (LLMs) [3, 8, 47] generate a sequence of tokens $T = (t_1, t_2, \cdots, t_n)$ to answer the question. Recent works like Chain-of-Thought (CoT) [50, 21] leverage prompt engineering in the context $C$ to encourage models to generate the intermediate reasoning process in $T$, which benefits LLM performance across diverse reasoning tasks. In this case, $T$ consists of a set of $m$ intermediate reasoning steps, which we denote as $S = (s_1, s_2, \cdots, s_m)$ . Each step $s_i$ can be represented by a subsequence of the generated tokens $(t_{l_i}, t_{r_i}) \subseteq T$. In much prior work, a generated solution is "correct" if and only if the predicted final answer in $s_m$ matches the ground truth $A$, which we call **answer correct(ness)**.

We observe that for all cases where LLMs produce erroneous final answers, there exists at least one mistake among the intermediate reasoning steps $S$. Moreover, even when the final answer is correct, there might still exist some mistakes among $S$. This phenomenon, as illustrated in Tab. 1, occurs for all LLMs we tested, including state-of-the-art models such as ChatGPT and GPT-4 [32]. Since later reasoning steps are conditioned on prior reasoning steps, these mistakes often initiate a snowball effect, causing subsequent mistakes to compound. This significantly diminishes the likelihood of correct problem-solving and impedes the progress towards achieving human-level complex reasoning.

Therefore, in this work, we place significant emphasis on ensuring the validity of *every* reasoning step, not just the correctness of the final answer. In particular, we focus on the validity of *deductive reasoning*, an essential component of a logical reasoning process. In deductive reasoning, we are

| Prompting | Reasoning Correctness | GSM8K | AQuA | MATH | AddSub | Date | Last Letters |
|---|---|---|---|---|---|---|---|
| Zero-shot | Correct | 98% | 96% | 100% | 98% | 98% | 100% |
| | Incorrect | 4% | 6% | 4% | 2% | 4% | 4% |
| | (Average) | 51% | 51% | 52% | 50% | 51% | 52% |
| Two-shot | Correct | 98% | 96% | 100% | 92% | 100% | 96% |
| | Incorrect | 2% | 4% | 0% | 6% | 26% | 6% |
| | (Average) | 50% | 50% | 50% | 49% | 63% | 51% |

Table 2: Zero-shot and two-shot reasoning chain verification accuracy for GPT-3.5-turbo (ChatGPT), where an entire reasoning chain is verified at once. The two shot prompt we used is presented in Appendix D.1. To generate verification inputs, for each dataset, we perform Chain-of-Thought (CoT) prompting and randomly sample 50 reasoning chains that are valid and 50 reasoning chains that exhibit mistakes. We observe that when given an *entire* reasoning process, where the deductive graphs for all reasoning steps are entangled together, it is challenging even for strong language models like ChatGPT to verify its validity.

given a (premise, conclusion) pair, and we are interested in determining whether the conclusion follows from the premises. In the context of reasoning-based QA tasks, for each reasoning step $s_i$, we define its *deductive validity* $V(s_i)$ as a binary variable. A reasoning step is **deductively valid** ($V(s_i) = 1$) if and only if $s_i$ can be logically deduced from its corresponding premises $p_i$, which consist of the context $C$, the question $Q$, and all the previous reasoning steps $s_j (j < i)$. Then, we can also define the deductive validity for the entire reasoning chain $S$ as $V(S) = \wedge_{i=1}^{M} V(s_i)$. Compared to evaluating answer correctness, which can be accomplished by simple functions such as exact string match, evaluating deductive validity is a lot more challenging. Thanks to the recent progress on LLMs, which demonstrate impressive in-context learning capabilities across diverse scenarios, we propose to use LLMs to examine reasoning chains and predict the deductive reasoning validity.

## 4 Deductively Verifiable Chain-of-Thought Reasoning

In this section, we introduce our specific approaches to performing deductive verification of reasoning chains. Specifically, we first introduce our motivation and method for decomposing a deductive verification process into a series of step-by-step processes, each only receiving contexts and premises that are necessary. Then, we propose **Natural Program**, a natural language-based deductive reasoning format, to facilitate local step-by-step verification. Finally, we show that by integrating deductive verification with unanimity-plurality voting, we can improve the trustworthiness of reasoning processes along with final answers. An overview of our approach is illustrated in Fig. 1 and Fig. 2.

### 4.1 Decomposition of Deductive Verification Process

Given a reasoning chain $S = (s_1, s_2, \cdots, s_n)$, a straightforward idea to verify its deductive validity is to ask LLMs to examine the *entire* reasoning chain at once. To assess the effectiveness of this approach, we conduct a preliminary experiment: for a dataset problem and its reasoning chain $S$ generated by ChatGPT, we prompt ChatGPT with "Do you think the above reasoning process is correct? Let's think step by step" such that its outputs whether there exists any mistake among any reasoning step in $S$. However, as demonstrated in Tab. 2, the verification accuracy is 50% for most datasets, and ChatGPT struggles at finding out mistaken reasonings. Notably, it persistently outputs "Correct" for most reasoning chain queries, regardless of their actual validity.

We conjecture that such phenomenon is caused by the abundance of irrelevant premises for each reasoning step. Recall that the premises $p_i$ for a reasoning step $s_i$ consist of the the question $Q$, the question context $C$, along with the prior reasoning steps $s_{\leq j} = \{s_j : j < i\}$. For $Q$ and $C$, we can further extract and decompose $Q \cup C$ into a set of "question-related premises" $QC = \{qc_1, qc_2, \cdots, qc_m\}$, where $qc_i$ is a premise or condition inferred from $Q \cup C$. Then, it is often the case that most elements of $p_i = QC \cup s_{\leq j}$ are irrelevant to the validity of $s_i$, leading to erroneous verifications from language models. A very recent work [41] also observes a similar phenomenon where LLMs are easily distracted by irrelevant context.

Hence, we propose a decomposition of the reasoning chain verification process into a series of step-by-step processes, where each step only considers the premises that are *necessary*. The overall validity

of the reasoning chain, denoted as $V(S) = \wedge_{i=1}^{M} V(s_i)$, can be naturally decomposed into individual step validity $V(s_i)$. However, achieving such decomposition is highly challenging without imposing constraints on the format of reasoning chains. Additionally, for each $s_i \in S$, we aim to ensure that it *explicitly* lists the minimal subset of premises $\bar{p}_i \subseteq p_i$ required for deductive reasoning to avoid potential ambiguities during verification. This motivates us to introduce a natural-language-based deductive reasoning format in Section 4.2.

## 4.2 Natural Program Deductive Reasoning Format

As previously mentioned in Sec. 4.1, we desire LLMs to output deductive reasoning processes that can be easily verified by themselves, specifically by listing out the minimal set of necessary premises $p_i$ at each reasoning step $s_i$. To accomplish its goal, we propose to leverage the power of natural language, which is capable of rigorously representing a large variety of reasoning processes and can be generated with minimal effort. In particular, we introduce **Natural Program** , a novel deductive reasoning format for LLMs. More formally, Natural Program consists of the following components:

- An instruction for models to extract question-related premises $QC$. We use the following instruction: "First, let's write down all the statements and relationships in the question with labels".

- A numbered-list of question-related premises, each prefixed with "#{premise_number}".

- An instruction for models to generate the reasoning chain $S$ based on the question-related premises $QC$. We use the following instruction: "Next, let's answer the question step by step with reference to the question and reasoning process".

- A list of prefixed reasoning steps $S_i$. The prefix has the following format: #{number} (by {list_of_premises_used}). Here "number" equals $|QC| + i$, and "list_of_premises_used" consists of numbers from the smallest subset of premises among $QC \cup s_{\leq j}$ that are used for the deductive reasoning of $s_i$. In addition, for the last reasoning step $s_m$, we ensure that it (1) includes a special tag Final Step; (2) refers to the premise number of the target question to be answered; (3) explicitly gives the final answer to a question.

To encourage language models to reason in the Natural Program format, we have designed one-shot prompts for different datasets, which are shown Appendix D.2. Given that LLM's reasoning outputs follow the Natural Program format, we can then verify the deductive validity of a *single* reasoning step $s_i$ through an instruction that consists of (1) the full descriptions of premises used for the reasoning of $s_i$; (2) the full description of $s_i$; (3) an instruction for validity verification, such as "Double-check the reasoning process, let's analyze its correctness, and end with "yes" or "no"." Note that throughout this verification process, we only retain the minimal necessary premise and context for $s_i$, thereby avoiding irrelevant context distraction and significantly improving the effectiveness of validation. Additionally, we employ a one-shot prompt for this verification process, which we find very helpful for improving the verification accuracy. The prompt is shown in Appendix D.3.

Figure 1 provides an overview of the complete Natural Program-based deductive reasoning and verification process. By using the Natural Program approach, we demonstrate that LLMs are capable of performing explicit, rigorous, and coherent deductive reasoning. Furthermore, Natural Program enables LLMs to self-verify their reasoning processes more effectively, enhancing the reliability and trustworthiness of the generated responses.

## 4.3 Integrating Deductive Verification with Unanimity-Plurality Voting

Given that we can *effectively* verify a deductive reasoning process, we can naturally integrate verification with LLM's sequence generation strategies to enhance the trustworthiness of both the intermediate reasoning steps and the final answers. In this work, we propose Unanimity-Plurality Voting, a 2-phase sequence generation strategy described as follows. Firstly, similar to prior work like [48], we sample $k$ reasoning chain candidates along with their final answers. In the unanimity phase, we perform deductive validation on each reasoning chain. Recall that a chain $S$ is valid (i.e., $V(S) = 1$) if and only if all of its intermediate reasoning steps are valid (i.e., $\forall i, V(s_i) = 1$). For *each* intermediate reasoning step $s_i$, we perform majority voting over $k'$ sampled single-step validity

predictions to determine its final validity $V(s_i)$. We then only retain the verified chain candidates $\{S : V(S) = 1\}$. In the plurality voting stage, we conduct a majority-based voting among the verified chain candidates to determine the final answer. This voting process ensures that the final answer is selected based on a consensus among the trustworthy reasoning chains.

## 5 Experiments

In this section, we perform evaluations to demonstrate the effectiveness of our Natural Program-based deductive reasoning verification approach over diverse reasoning datasets. Firstly, we show that our deductive verification process leads to substantial improvements in the rigor and reliability of reasoning chains. Subsequently, we will examine the impact of deductive verification on the accuracy of final answers. Our findings reveal that by adopting our Natural Program reasoning format without verification, we improve answer correctness on challenging benchmarks. Further applying deductive verification leads to slight reductions in final answer accuracy. One reason for this phenomenon is that the verification process effectively identifies and eliminates flawed reasoning chains that still produce correct answers.

### 5.1 Experimental Setup

**Benchmarks.** We evaluate the deductive verification accuracy and the answer correctness of reasoning chains over a diverse set of reasoning tasks: arithmetic reasoning, symbol manipulation, and date understanding. For arithmetic reasoning, we utilize the following benchmarks: 1) AddSub [19]; 2) GSM8K [10]; 3) MATH [17]; 4) AQuA [24]. Among these benchmarks, the AddSub and GSM8K datasets involve middle school-level multi-step calculations to arrive at a single number as the final answer. The MATH dataset presents more challenging problems that require expressing the answer as a mathematical expression in LaTeX format. These problems involve concepts from linear algebra, algebra, geometry, calculus, statistics, and number theory. AQuA also features similarly challenging problems, except that questions are in a multiple-choice format. For symbol manipulation, we use Last Letter Concatenation [50], where the model is tasked with concatenate the last letters of all the words provided in the question. For date understanding, we use the one from BIG-bench [45]

**Deductive verfication evaluation setup.** For each of the above benchmarks, we select 100 reasoning chains, where 50 of them are deductively valid and 50 of them exhibit reasoning mistakes. The ground-truth deductive validity of each reasoning chain is determined by human annotators.

**Answer extraction.** To extract answers from reasoning solutions, we first perform text splitting based on answer prefix patterns such as "answer is" or "option is". Then, using problem type-specific regular expressions, we extract the final answer. To extract the validity results from deductive verification processes, we only keep the last sentence of model response. We then extract the validity answer with regular expressions to obtain attitude words, e.g., "yes" or "no", to determine the validity answer. Sometimes, language models may not provide a direct answer and instead output phrases like "not applicable" at the end of the response. In such cases, we consider the answer from the model as "yes". Please refer to Appendix C for more details.

**Model and Hyperparameters.** We conduct our main experiments with GPT-3.5-turbo (Chat-GPT) [32]. We also present results for the LLama model-family [47]) in Appendix A, where we find the deductive verification accuracy to be worse than larger models even after finetuning. For ChatGPT, we use a generation temperature of $T = 0.7$. For Unanimity-Plurality Voting, we set $k = 10$ and $k' = 3$ by default. We use 1-shot prompting for both reasoning chain generation and deductive verification (except reasoning chain generation for the date understanding task where we use 2-shot). See Appendix D.2 and Appendix D.3 for more details.

### 5.2 Comparison of Deductive Verification Accuracy

We compare the verification accuracy of reasoning chains using two methods: (1) verifying the entire reasoning chain at once (as described in Section 4.1) without utilizing the Natural Program, and

---

[1] Most results for Faithful CoT are from their official repository `https://github.com/veronica320/Faithful-CoT`, except MATH and AddSub due to the unavailability. For these two datasets, we use our implementation and the same prompt for the math word problems in their paper. The prompt for Last Letters is not available, so we leave it blank.

| Verification Method | Reasoning Correctness | GSM8k | AQuA | MATH | AddSub | Date | Last Letters | Overall |
|---|---|---|---|---|---|---|---|---|
| CoT Two-shot | Correct | 98% | 96% | 100% | 92% | 100% | 96% | 97% |
| | Incorrect | 2% | 4% | 0% | 6% | 26% | 6% | 7% |
| | (Average) | 50% | 50% | 50% | 49% | 63% | 51% | 52% |
| Natural Program One-shot | Correct | 84% | 72% | 70% | 95% | 90% | 96% | 85% |
| | Incorrect | 84% | 62% | 76% | 40% | 56% | 6% | 54% |
| | (Average) | **84%** | **67%** | **73%** | **68%** | **73%** | 51% | **69%** |

Table 3: Comparison of deductive verification accuracy of reasoning chains for GPT-3.5-turbo (Chat-GPT). We compare two approaches: (1) verifying entire reasoning chains generated by Chain-of-Thought prompting; (2) verifying reasoning chains generated in the Natural Program format with step-by-step decomposition. In the latter case, when we verify each reasoning step $s_i$, we only keep the necessary subset of premises $\bar{p}_i \subseteq p_i$. To calculate verification accuracy, for each dataset, we randomly sample 50 reasoning chains that are deductively valid and 50 reasoning steps exhibiting incorrect reasonings.

| Methods | Arithmetic | | | | Commonsense | |
|---|---|---|---|---|---|---|
| | GSM8K | AQuA | MATH* | AddSub | Date | Last Letters |
| CoT + Voting | **87.62%** | 70.18% | 35.93% | 92.36% | 69.97% | 81.60% |
| Faithful CoT + Voting | 75.80% | 61.80% | 31.78%[1] | 88.35%[1] | **73.50%** | - |
| Ours (Natural Program (NP), No Verification) | 87.05% | **70.34%** | **36.75%** | **93.67%** | 72.49% | **92.98%** |
| Ours (NP + Deductive Verification + UPV) | 86.01% | 69.49% | 36.48% | 93.54% | 71.45% | 92.60% |

Table 4: Final answer accuracy comparison on GPT-3.5-turbo (ChatGPT). All approaches generate $k = 10$ reasoning chains for each problem before performing majority voting or reasoning chain filtering with our deductive verification approach.

(2) our Natural Program-based verification approach with step-by-step decomposition. The results, presented in Table 3, indicate that our approach achieves significantly higher reasoning verification accuracy across most datasets. It effectively identifies erroneous reasoning in faulty chains while maintaining a low rate of false positives for valid chains. However, we observe that our approach's effectiveness is limited on the "Last Letters" task. We hypothesize that this is due to the task's nature, where each subsequent reasoning step is conditioned on *all* previous steps, presenting greater challenges for reasoning verification due to the increased dependency among premises.

### 5.3 Impact of Natural Program and Deductive Verification on Final Answer Correctness

We then investigate the impact of our Natural Program reasoning format and our deductive verification process on final answer correctness. We conduct two experiments: (1) for each problem, we instruct language models to generate $k = 10$ reasoning chain candidates in the Natural Program (NP) format and perform simple majority voting on final answers, *without* using deductive verification to filter out reasoning chain candidates; (2) applying our deductive verification approach to filter out reasoning chain candidates, and apply Unanimity-Plurality Voting (UPV) along the process to determine the final answer. As a reference, we also report the performance of Chain-of-Thought (CoT) [50] and Faithful CoT [27]. For these baselines, we perform simple answer-based majority voting with $k = 10$ for fair comparison.

Results are presented in Tab. 4. While our major goal is to improve the trustworthiness and reliability of deductive reasoning, we find that prompting language models to reason in our Natural Program format achieves on-par or better final answer accuracy than baselines over many reasoning tasks. Upon further applying our deductive verification approach to filter out invalid reasoning chains, we observe a slight decrease in final answer accuracy. One major contributing factor to this decrease is the filtering out of reasoning chain candidates that provide correct answers but exhibit incorrect reasoning. We illustrate an example in Table 5, where ChatGPT generates the correct final answer but assigns incorrect premise numbers to support the first reasoning step. We note that in many such cases, our approach effectively identifies these reasoning errors, thereby enhancing the rigor and

| | |
|---|---|
| **Question**: In March it rained 0.81 inches. It rained 0.35 inches less in April than in March. How much did it rain in April? | |
| **Ground Truth:** It rained 0.81 - 0.35 = 0.46 inches. | |
| **GPT-3.5-turbo (ChatGPT):**
**Question-related premises:**
#1. It rained 0.81 inches in March.
#2. It rained 0.35 inches less in April than in March.
#3. How much did it rain in April?
**The first reasoning step:**
#4. (by #2) Step 1: Calculate how much it rained in April by subtracting the difference of 0.35 inches from the amount in March.
Amount of rain in April:  0.81 - 0.35 = 0.46 inches | |

Table 5: For this reasoning chain candidate of this GSM8k problem, ChatGPT generates the correct final answer but provides incorrect premise numbers for grounding the first reasoning step. In "Step 1", the correct premise numbers should be #1 and #2. Our deductive reasoning verification approach effectively identifies these reasoning errors, enhancing the rigor and trustworthiness of the reasoning process. It is worth noting that removing a correct answer from the candidate reasoning chains has a slightly negative impact on the overall final answer correctness.

| Premise Context | # Shots | Reasoning Correctness | GSM8K | AQuA | MATH | AddSub | Date | Last Letters | Average |
|---|---|---|---|---|---|---|---|---|---|
| Full Premises | 1 | Correct | 64% | 54% | 58% | 95% | 26% | 96% | 66% |
| | | Wrong | 56% | 68% | 56% | 24% | 76% | 5% | 48% |
| | | (Average) | 60% | 61% | 57% | 60% | 51% | 51% | 57% |
| Minimal Premises | 0 | Correct | 84% | 78% | 90% | 96% | 90% | 12% | 75% |
| | | Wrong | 26% | 12% | 28% | 20% | 20% | 80% | 31% |
| | | (Average) | 55% | 45% | 59% | 58% | 55% | 46% | 53% |
| Minimal Premises | 1 | Correct | 84% | 72% | 70% | 95% | 90% | 96% | 85% |
| | | Wrong | 84% | 62% | 76% | 40% | 56% | 6% | 54% |
| | | (Average) | **84%** | **67%** | **73%** | **68%** | **73%** | 51% | **69%** |

Table 6: Ablation study on the impact of (1) premise context and (2) zero-shot vs. few-shot scenarios on deductive verification accuracy using our Natural Program-based approach with step-by-step reasoning chain decomposition. To verify each reasoning step $s_i$, we either the full premises $p_i = QC \cup S_{\leq j}$, or use the minimal subset of premises $\bar{p}_i \subseteq p_i$ necessary as outlined in Sec. 4.1 The one-shot prompt we used is shown in Appendix D.3. For each dataset, we randomly sample 50 reasoning chains that are deductively valid and 50 reasoning steps exhibiting incorrect reasonings.

reliability of the language models' reasoning processes, albeit with a slight negative impact on the overall final answer correctness. Further discussions are presented in Appendix B.

## 5.4 Ablation Study

In addition, we perform several ablation studies to gain further insights into the designs of our deductive verification approach. In Tab. 6, we compare two different approaches to verify a single reasoning step $s_i \in S$ following our Natural Program format. The first approach utilizes all premises $p_i = QC \cup S_{\leq j}$ for verification regardless of their relevance to $s_i$, potentially introducing irrelevant contexts. The second approach follows our design in Sec. 4.1 and only includes the necessary context and premises $\bar{p}_i \subseteq p_i$. We observe that removing irrelevant premises significantly improves the reasoning chain verification accuracy on many datasets, highlighting the importance of this technique.

We also ablate on our Unanimity-Plurality Voting strategy by investigating the impact of different $k'$. Recall that $k'$ determines the number of votes to produce validity predictions of single-step reasoning. Results are shown in Tab. 7. We observe that increasing $k'$ generally enhances reasoning validation accuracy, though we note that this is at the expense of more compute.

## 6  Limitations

While we have demonstrated the effectiveness of Natural Program-based deductive reasoning verification to enhance the trustworthiness and interpretability of reasoning steps and final answers, it is

| Answer Correctness | $k' = 1$ | $k' = 3$ | $k' = 5$ | $k' = 10$ |
|---|---|---|---|---|
| Correct | 86% | 90% | 90% | 92% |
| Wrong | 38% | 38% | 38% | 40% |

Table 7: Ablation of different values of $k'$ on the verification accuracy of reasoning chains using our Unanimity-Plurality Voting strategy. Experiments are performed on AddSub using GPT-3.5-turbo (ChatGPT).

---

**Question**: Melanie had 10 quarters and 17 pennies in her bank. Her dad gave her 27 pennies and her mother gave her 19 pennies. How many pennies does Melanie have now?
**Ground Truth**: Melanie have 17 + 27 + 19 = 63 pennies.

**ChatGPT's reasoning step:**
#5. (by #1) Step 1: Calculate the number of pennies Melanie had initially.
Number of pennies in 10 quarters: 10 * 25 = 250
Number of pennies initially: 250 + 17 = 267

---

Table 8: An example question with ambiguous wordings. The term "pennies" in this question can be interpreted as either a type of coin or a unit of currency. In this particular question, "pennies" is treated as a type of coin. However, the initial reasoning step by ChatGPT mistakenly treats "pennies" as a unit of currency, resulting in the conversion of all Melanie's money into "pennies" (highlighted in red). Consequently, all subsequent reasoning steps follow this flawed logic, leading to an incorrect reasoning trace. Our deductive verification is not yet able to detect such errors.

important to acknowledge that our approach has limitations. In this section, we analyze a common source of failure cases to gain deeper insights into the behaviors of our approach. The failure case, as shown in Tab. 8, involves the ambiguous interpretation of the term "pennies," which can be understood as either a type of coin or a unit of currency depending on the context. The ground truth answer interprets "pennies" as coins, while ChatGPT interprets it as a unit of currency. In this case, our deductive verification process is incapable of finding such misinterpretations. Contextual ambiguities like this are common in real-world scenarios, highlighting the current limitation of our approach.

## 7   Conclusion

In this paper, we aim to enable Large Language Models (LLMs) to perform explicit and rigorous deductive reasoning while ensuring the trustworthiness of their reasoning processes through self-verification. To this end, we have proposed a novel framework based on "Natural Program", a natural language-based deductive reasoning format that facilitates reasoning verification and can be easily generated through in-context learning. Within this framework, we decompose the verification process of complex reasoning chains into step-by-step subprocesses that focus solely on necessary context and premises, allowing us to significantly enhance the accuracy of verification. Additionally, we introduce a Unanimity-Plurality Voting strategy to further improve verification accuracy. Experimentally, we demonstrate the superiority of our framework in improving the rigor, trustworthiness, and interpretability of reasoning steps and answers.

**Broader Impact.** While our deductive verification approach can mitigate hallucinations and reasoning errors of Large Language Models (LLMs), it does not completely eliminate these phenomena. LLMs can still produce harmful and biased content, make incorrect claims, and produce wrongful advice. This issue becomes particularly significant when LLMs engage in complex reasoning chains, increasing the risk of misleading users. Consequently, it is still crucial for users to exercise great caution when interacting with, deploying, or developing LLM-based applications.

## Acknowledgements

We would like to express our sincere gratitude to Tongzhou Mu and Caiwei Xiao from UC San Diego, Kairong Luo from Tsinghua University, and Pulkit Madan, Reza Pourreza, Sunny Panchal, and Apratim Bhattacharyya from Qualcomm for their valuable discussions and feedback.

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

# A   Deductive Verification with Vicuna Models

We further explore the efficacy of deductive verification for open-source models. We select two popular models: Vicuna-7B and Vicuna-13B [7]. These models are fine-tuned versions of LLaMA-7B and LLaMA-13B [47] using the ShareGPT data[3]. We use the same Natural Program-based one-shot verification method we used in the main paper. Results are shown in the first and the third rows of Table 9. We observe for the *original Vicuna models without finetuning*, Vicuna-7B exhibits poor performance in deductive verification and fails to find out reasoning mistakes, while the larger Vicuna-13B exhibits better verification accuracy.

| Models | Reasoning Correctness | GSM8K | AQuA | MATH | AddSub | Date | Last Letters | Overall |
|---|---|---|---|---|---|---|---|---|
| Vicuna-7B | Correct | 80% | 86% | 96% | 98% | 96% | 80% | 89% |
| | Wrong | 14% | 22% | 16% | 6% | 20% | 34% | 19% |
| | (Average) | 47% | 54% | 56% | 52% | 58% | 57% | 54% |
| Vicuna-7B (fine-tuned) | Correct | 68% | 48% | 46% | 76% | 46% | 32% | 53% |
| | Wrong | 72% | 86% | 54% | 60% | 72% | 68% | 69% |
| | (Average) | 70% | **67%** | 50% | 68% | 61% | 50% | 61% |
| Vicuna-13B | Correct | 86% | 82% | 92% | 96% | 72% | 74% | 84% |
| | Wrong | 32% | 36% | 20% | 20% | 34% | 30% | 29% |
| | (Average) | 59% | 59% | 56% | 58% | 53% | 52% | 57% |
| Vicuna-13B (fine-tuned) | Correct | 74% | 50% | 56% | 86% | 72% | 12% | 58% |
| | Wrong | 72% | 76% | 72% | 68% | 62% | 96% | 74% |
| | (Average) | **73%** | 63% | **64%** | **77%** | **67%** | **54%** | **66%** |
| ChatGPT (GPT-3.5-Turbo) | Correct | 84% | 72% | 70% | 95% | 90% | 96% | 85% |
| | Wrong | 84% | 62% | 76% | 40% | 56% | 6% | 54% |
| | (Average) | 84% | 67% | 73% | 68% | 73% | 51% | 69% |

Table 9: One-shot Deductive Verification Accuracy of Vicuna-7B and Vicuna-13B. The models are evaluated with or without finetuning on our deductive verification dataset. For each dataset, we randomly sample 50 reasoning chains that are deductively valid and 50 reasoning steps exhibiting incorrect reasonings.

We therefore conduct an additional experiment to investigate if the verification accuracy of Vicuna models can be improved by fine-tuning. To this end, we generate a deductive verification dataset, which consists of 2000 reasoning steps evenly distributed between correct and incorrect categories. We automatically generate this dataset using GPT-3.5-turbo since it exhibits a very high accuracy of single-step verification. We first use GPT-3.5-turbo to generate solutions for problems in GSM8K's training set. We then execute step-by-step deductive verification on these solutions using GPT-3.5-turbo. For solutions that result in correct final answers, we retain the reasoning steps that pass deductive verification. For solutions that yield incorrect final answers, we retain the reasoning steps that cannot pass deductive verification. After constructing our dataset, we then fine-tune the Vicuna models using the verifications of the 2000 reasoning steps. Models were fine-tuned with 4 A100-80GB over 3 epochs. Training parameters are shown in Table 10.

As shown in Tab. 9, we observe that fine-tuning with our dataset can enhance the deductive verification accuracy of Vicuna models not only on the dataset where the training dataset is constructed (GSM8K), but also on many other datasets. However, the accuracy is still worse than non-finetuned GPT-3.5, which suggests that model capacity has a significant impact on deductive verification capabilities.

# B   More Discussion on Improvements of Deductive Verification Accuracy Versus Improvements on Final Answer Correctness

In the main paper, we demonstrated that our verification approach significantly improves the verification accuracy of reasoning chains (Tab. 3, 6, but barely improves the final answer accuracy (Tab. 4). We further analyze this phenomenon below:

---

[3]https://github.com/domeccleston/sharegpt

| Hyperparameters | Value |
|---|---|
| Optimizer | AdamW |
| Learning rate | $1 \times 10^{-5}$ |
| Weight decay | 0.00 |
| Num epochs | 3 |
| Batch size | 64 |
| Learning rate schedule | Linear |

Table 10: Hyperparameters for finetuning Vicuna models with our deductive verification dataset.

Consider the GSM8K dataset as an example (recall that the final answer for a problem is obtained through majority voting). Among all problems, 91.6% of problems have |(number of votes received by the correct answer) − (largest number of votes received by a single wrong answer)| > 2, and their final answers are unlikely to be changed through our deductive verification approach. For the rest of the cases (8.4%), where deductive verification is more likely to impact their final answers, we found that:

- Among all reasoning chains that arrive at correct answers (these correct-answer chains account for 49.4% of all reasoning chain candidates), 46.2% of reasoning chains are filtered out by our verification process.

- Among the reasoning chains that arrive at correct answer but are filtered out by our verification process, 76.3% indeed exhibit incorrect reasoning.

- Among the reasoning chains that arrive at correct answer and are not filtered out by our verification process, 78.0% indeed have correct reasonings.

- Among the reasoning chains that do not arrive at correct answer and exhibit incorrect reasonings (these account for 50.6% of all reasoning chain candidates), 40.6% are filtered out by our verification process.

The above statistics shows that a significant portion of reasoning chains that arrive at correct answers but exhibit incorrect reasoning are successfully eliminated. Therefore, the reliability and trustfulness of reasoning chains that arrive at the correct answers are significantly improved. Combined with the fact that a significant proportion of reasoning chains that exhibit incorrect answers are eliminated, and that our approach's verification accuracy significantly improves over naive verification approaches, our primary goal to improve LLM reasoning reliability is accomplished.

Nevertheless, the removals of many reasoning chains yielding correct answers (specifically, a significant 46.2% × 49.4% of all chains) has a notable impact. This even exceeds the removals of reasoning chains with incorrect reasonings and answers (40.6% × 50.6% of all chains). As a result, there are fewer votes for the correct answer when generating final answers through majority voting, which limits the final answer accuracy. In the future, we believe that when a greater proportion of incorrect reasoning chains with incorrect answers are filtered out, we can improve the final answer accuracy.

## C  More Details on Answer Extraction

In this section, we describe our process to extract the final answer from language models' responses. The process begins by selecting the last three non-empty lines. Then, these lines are processed through the following pipeline:

1. Firstly, we use a list of regular expressions to identify "No-Answer" patterns within the text, such as "we cannot answer (this|the) question". This process helps us ascertain whether the model can provide a conclusive answer. If any such patterns appear in the text, we mark "No answer!" as the final answer. However, if we don't detect these patterns, we proceed to the next steps for extracting the final answer.

2. Secondly, if any "Answer-Split" patterns are found in the text, we divide the text into several blocks using the identified pattern. The last block of text is then utilized for extracting the answer.

3. Lastly, we use regular expressions, as outlined in Tab. 11, to scan the remaining text for possible final answers. If multiple matches are found for the pattern, we select the first match as the final answer. If no pattern matches are found in the remaining text, we default the final response to "No answer!".

**"No-Answer" Patterns:** "we cannot provide an answer to this question with (this|the) given information", "we cannot answer (this|the) question", "we cannot determine", "we can't determine", "we do not have enough information to answer (this|the) question", "we do not have enough information to provide a definitive answer to (this|the) question", "the answer(.*?)is unknown", "answer is not listed among the answer choices".

**"Answer-Split" Patterns:** "answer is", "final answer:", "answer to the question is", "answer to this question is", "concatenated letters are", "concatenate the letters -", "The answer of ".

| Answer Type | Regular Expression |
|---|---|
| Number | (-?\d[\d,\. ]*) |
| Fractional number | (-?\(\d+\/\d+\)\/\d+\|-?\d+\/\d+) |
| Date | (\d\d\/\d\d\/\d\d\d\d) |
| Yes or No | (?:Yes\|No\|yes\|no\|NO\|YES) |

Table 11: Regular Expression for extracting the final answers of different kinds of questions.

## D Prompts

### D.1 Prompt for Direct Reasoning Chain Verification Without Natural Program Format

For the results in Tab. 2 of the main paper, We use "Do you think the above reasoning process is correct? Let's think step by step." as the zero-shot prompt to verify an entire reasoning chain at once. We also design a two-shot prompt for reasoning chain verification as shown in Tab. 12, which covers one correct reasoning chain and one incorrect reasoning chain.

### D.2 Prompts for Reasoning Chain Generation in the Natural Program Format

To instruct models to generate reasoning chains in the Natural Program format that facilitates step-by-step deductive verification, we have designed four distinct prompts to address different types of problems. These include:

1. Math word problems, as illustrated in Tab. 13, covering GSM8K, MATH, and AddSub datasets.
2. Math word problems with multiple-choice options, illustrated in Tab. 14, covering the AQuA dataset.
3. Date-related problems, illustrated in Tab. 15, covering the Date dataset.
4. Last Letters problems, illustrated in Tab. 16, covering the Last Letters dataset.

### D.3 Prompt for Deductive Verification Following Natural Program Format and Step-by-Step Decomposition

We have designed a general one-shot prompt for the deductive verification of a single reasoning step on different datasets, as shown in Tab. 17. This prompt serves to instruct language models to generate the deductive validity of each reasoning step as illustrated in Sec. 4.2 and the top-right box of Fig. 1 of the main paper.

## E More Deductive Verification Examples

In this section, we present more deductive verification examples using our Natural Program-based approach on single reasoning steps.

In Tab. 18, we demonstrate that the language model (ChatGPT) not only successfully identifies ungrounded information, but also identifies logical errors within the given solutions.

In Tab. 19, we illustrate a case where the language model fails to detect ungrounded premise numbers, mistakenly assuming that these numbers can be derived from grounded ones.

Lastly, in Tab. 20, we illustrate a case where the language model is sometimes unable to correctly identify grounded numbers.

**Here is a math question and its solution:**
"Question:
Jason had 20 lollipops. He gave Denny some lollipops. Now Jason has 12 lollipops. How many lollipops did Jason give to Denny?

Answer:
Jason started with 20 lollipops. Then he had 12 after giving some to Denny. So he gave Denny 20 + 12 =34. The answer is 34."

You are a math teacher. Do you think the reasoning process is correct?
Let's check the "Answer" step by step, and then end with "yes" or "no".

**Answer:**
The problem statement is clear, but the reasoning process has a major error in it.
Let's go through it step by step:

1. "Jason started with 20 lollipops." This is correct according to the problem statement.

2. "Then he had 12 after giving some to Denny." This is also correct as per the problem.

3. "So he gave Denny 20 + 12 =34. The answer is 34." This is incorrect. The correct reasoning process should be subtracting the number of lollipops he ended up with from the number he started with. That would be 20 - 12 = 8 lollipops.

So, Jason gave Denny 8 lollipops, not 34. No, the reasoning process is incorrect.

**Here is a math question and its solution:**
"Question:
Olivia has $23. She bought five bagels for $3 each. How much money does she have left?

Answer:
Olivia had 23 dollars. 5 bagels for 3 dollars each will be 5 x 3 = 15 dollars. So she has 23 - 15 dollars left. 23 - 15 is 8. The answer is 8."

You are a math teacher. Do you think the reasoning process is correct?
Let's check the "Answer" step by step, and then end with "yes" or "no".

**Answer:**
Let's go through this reasoning process step by step.

1. "Olivia had 23 dollars." This is the given starting amount of money that Olivia has. This step is correct.

2. "5 bagels for 3 dollars each will be 5 x 3 = 15 dollars." This is correct as well. Olivia spends 3 dollars for each bagel, and she buys 5 of them. So the total cost of the bagels is 3 dollars/bagel x 5 bagels = 15 dollars.

3. "So she has 23 - 15 dollars left." This statement is also correct. After purchasing the bagels, Olivia will have her initial amount of money minus the cost of the bagels.

4. "23 - 15 is 8. The answer is 8." This step is correct as well. 23 dollars (Olivia's initial amount) - 15 dollars (the cost of the bagels) = 8 dollars. This means that Olivia has 8 dollars left after buying the bagels.

So overall, Olivia has 8 dollars left after buying five bagels for 3 dollars each. Yes, the reasoning process is correct.

Table 12: Two-shot prompt for direct reasoning chain verification without Natural Program format.

**Question:**
Alice buys 5 apples. Each apple costs her $2. She gets them for 50% off. After she bought she decides she doesn't like 2 of them and sells them to her friends for 1. How much money was she out?

**Answer:**
First, let's write down all the statements and relationships in the question with labels.
#1. Alice gets 5 apples.
#2. Each apple costs $2, but Alice gets apples for 50% off.
#3. Alice sells 2 of 5 apples for 1.
#4. How much money was Alice out?

Next, let's answer the question step by step with reference to the question and reasoning process:

#5. (by #2) Step 1: Calculate the price per apple after a 50% discount.
Original price per apple: $2
Discount: 50%
Price per apple after discount: $2 * (1 - 0.50) = $2 * 0.50 = $1

#6. (by #1 #5) Step 2: Calculate Alice's cost of the 5 apples after a 50% discount.
Price per apple after discount: $1
Total apples: 5
Alice's total cost of 5 apples: $1 * 5 = $5

#7. (by #3) Step 3: Calculate the money Alice gets back from selling 2 apples.
Money that Alice gets back from selling 2 apples: $1

#8. (by #6 #7) Step 4: Calculate the total amount of money Alice is out.
Alice's total cost of 5 apples: $5
Money that Alice gets back from selling 2 apples: $1
Money that Alice is out: $5 - $1 = $4

#9. (by #4 #8) The original question is #4.
How much money was Alice out? We do not miss information on the rewritten labels. So the answer to this question is Alice is out $4.

Please answer the following question by following the above template.

Table 13: One-shot Natural Program prompt for reasoning chain *generation* on math word problems.

**Question:**
Alice buys 5 apples. Each apple costs her $2.
She gets them for 50% off. After she bought She decides she doesn't like 2 of them and sells them to her friends for 1.
How much money was she out?
Answer Choices:
A) 4
B) 2
C) None of the above

**Answer:**
First, let's write down all the statements and relationships in the question with labels.
#1. Alice gets 5 apples.
#2. Each apple costs $2, but Alice gets apples for 50% off.
#3. Alice sells 2 of 5 apples for 1.
#4. How much money was Alice out?

Next, let's answer the question step by step with reference to the question and reasoning process:

#5. (by #2) Step 1: Calculate the price per apple after a 50% discount.
Original price per apple: $2
Discount: 50%
Price per apple after discount: $2 * (1 - 0.50) = $2 * 0.50 = $1

#6. (by #1 #5) Step 2: Calculate Alice's cost of the 5 apples after a 50% discount.
Price per apple after discount: $1
Total apples: 5
Alice's total cost of 5 apples: $1 * 5 = $5

#7. (by #3) Step 3: Calculate the money Alice gets back from selling 2 apples.
Money that Alice gets back from selling 2 apples: $1

#8. (by #6 #7) Step 4: Calculate the total amount of money Alice is out.
Alice's total cost of 5 apples: $5
Money that Alice gets back from selling 2 apples: $1
Money that Alice is out: $5 - $1 = $4

#9. (by #4 #8) The original question is #4. How much money was Alice out? We do not miss information on the rewritten labels. So the answer to this question is Alice is out $4. Among all the answer choices, the best option is A) 4.

Please answer the following question by following the above template.

Table 14: One-shot Natural Program prompt for reasoning chain generation on math word problems with multiple choice.

Let's work on date computation.
**Question:**
Yesterday is 02/22/2012. What is the date one week from today? Please answer in format MM/DD/YYYY.

**Answer:**
First, let's write down all the statements and relationships in the question with labels.
#1. Yesterday is 02/22/2012.
#2. What is the date one week from today? Please answer in format MM/DD/YYYY.

Next, let's answer the question step by step with reference to the question and reasoning process:

#4. (by #1) Step 1: Calculate the date today.
The date of Yesterday is 02/22/2012.
The date of today is 02/23/2012.

#5. (by #2 #4) Step 2: Calculate the date one week from today and write the answer in MM/DD/YYYY.
Because 2012 is a leap year and February in 2012 has 29 days, the date one week from today is 03/01/2012.
Therefore, date one week from today is 03/01/2012.

**Question:**
Today is 3/2 and Bob is watching NBA 2007. What is the date one week before today?
Please answer in format MM/DD/YYYY.

**Answer:**
First, let's write down all the statements and relationships in the question with labels.
#1. Today is 3/2.
#2. Bob is watching NBA 2007 today.
#3. What is the date one week before today? Please answer in format MM/DD/YYYY.

Next, let's answer the question step by step with reference to the question and reasoning process:

#4. (by #1 #2) Step 1: Calculate the date today.
Today is 3/2 and Bob is watching NBA 2007 today. Thus, Today is 03/02/2007.

#5. (by #3 #4) Step 2: Calculate the date one week before today and write the answer in MM/DD/YYYY.
Because 2007 is not a leap year and February in 2007 has 28 days, the date one week from today is 02/23/2007.
Therefore, the date one week before today is 02/23/2007.

Please answer the following question by following the above template.

Table 15: Two-shot Natural Program prompt for reasoning chain generation on the Date dataset.

Let's work on date computation.
**Question:**
Take the last letters of each words in "Apple Is" and concatenate them.

**Answer:**
First, let's write down all the statements and relationships in the question with labels.

#1. We need to take the last letter of the word "Apple".
#2. We need to take the last letter of the word "Is".
#3. We need to concatenate the last letters of the words sequentially

Next, let's answer the question step by step with the complete reference to the question and reasoning process:

#4. (by #1, #3) Step 1: Taking the last letter of the word "Apple" and concatenate the latter.
The last letter of the word "Apple" is "e".
The concatenated last letters are "e".

#5. (by #2, #3, #4) Step 2: Taking the last letter of the word "Is" and concatenate the latter.
The last letter of the word "Is" is "s".
The concatenated last letters are "es".

#6. (by #3, #5) Step 3: Provide the final answer of the concated letter.
Therefore, the answer is "es".

Please answer the following question by following the above template.

Table 16: One-shot Natural Program prompt for reasoning chain generation on the Last Letters dataset.

**Here is some information:**
"1. Alice gets 2 apples.
2. Alice gets twice as many oranges as bananas.
3. Original price per apple: \$2
Discount: 50%
Price per apple after discount: \$2 * (1 - 0.50) = \$2 * 0.50 = \$1"

Based on the given information, here is a reasoning process:
"Calculate Alice's cost of the 5 apples after a 50% discount.
Price per apple after discount: \$1
Apples be eaten: 3
Oranges be eaten: 6
Total apples: 5
Total oranges: x
Total bananas: 2x
Alice's total number of fruits: 5 + x + 2x = 5 + 4x
Alice's total cost of 5 apples: \$1 * (5 - 3) = \$2"

Double-check the reasoning process, let's analyze its correctness, and end with "yes" or "no".

**Answer:**
Let's think step by step.

Grounding check
Purpose: Find references for numbers in the reasoning process sequentially
Rule: Any ungrounded number makes the check fails

"x" is a variable, we do not care about variables.
"y" is a variable, we do not care about variables.

"5" is a quantity from the given information that represents the number of apples.
"50%" is a quantity from the given information that represents the discount.
"\$1" is a quantity from the given information that represents the price per apple after the discount.
"3" is a quantity not from the given information, it is ungrounded.
"6" is a quantity not from the given information, it is ungrounded.
"\$2" is a quantity from the given information that represents the original price per apple.
Result: Based on the check, there are ungrounded numbers, so the grounding check fails.

Reasoning check:
Purpose: Check whether the reasoning logic correctly answers the question
Rule: Any logic error makes the check fails

To calculate the total cost of apples after a 50% discount, we should multiply the number of apples by the discounted price. But to answer the question, we do not need to calculate the total number of fruit Alice gets.
Result: Based on the check, there are some logic mistakes, so the reasoning check fails.

Calculation check:
Purpose: Check whether the calculation process is consistent
Rule: Any inconsistent calculation makes the check fails

calculation1:
equation: \$1 * (5 - 3), answer: \$2
(5 - 3) = 2
\$1 * 2 = \$2 is consistent with the answer, so the calculation is correct.
calculation2:
equation: 5 + x + 2x, answer: 5 + 4x
x + 2x = 3x
5 + 3x is inconsistent with the answer, so the calculation is incorrect.
Result: Based on the check, the calculation process is inconsistent, so the calculation check fails.

Check results: Ground check fails, Reasoning check fails, Calculation check fails.
Rule: Any failed check makes the reasoning incorrect.
So the answer is "no".

Table 17: One-shot prompt for deductive verification of a single reasoning step, following our Natural Program format and step-by-step reasoning chain decomposition.

**Input:**
Here is some information:
"For every 25 telephone calls he makes, he gets one person to come into the dealership."

Based on the given information, here is a reasoning process:
"Calculate the number of people that would come into the dealership for 15 car sales.
Number of people needed to sell 15 cars: 15 * 2 = 30"

Double-check the reasoning process, let's analyze its correctness, and end with "yes" or "no".

Answer:
Let's think step by step without any assumptions.

---

**Output:**
Grounding check:
Purpose: Find references for numbers in the reasoning process sequentially
Rule: Any ungrounded number makes the check fails

"15" is a quantity from the question that represents the number of car sales.
"2" is a quantity not from the given information, it is ungrounded.
"25" is a quantity from the given information that represents the number of telephone calls
per one person to come into the dealership.
Result: Based on the check, there are ungrounded numbers, so the grounding check fails.

Reasoning check:
Purpose: Check whether the reasoning logic correctly answers the question
Rule: Any logic error makes the check fails

It is not clear how the number of people coming into the dealership relates to the number of car sales.
The given information only tells us the ratio of telephone calls to people coming into the dealership,
but it does not provide any information about the number of telephone calls made for each car sale.
Therefore, the reasoning process is not correct.
Result: Based on the check, there is a logic mistake, so the reasoning check fails.

Calculation check:
Purpose: Check whether the calculation process is consistent
Rule: Any inconsistent calculation makes the check fails

calculation1:
equation: 15 * 2, answer: 30
15 * 2 = 30 is consistent with the answer, so the calculation is correct.
Result: Based on the check, the calculation process is consistent, but the reasoning is incorrect,
so the calculation check fails.

Check results: Grounding check fails, Reasoning check fails, Calculation check fails.
Rule: Any failed check makes the reasoning incorrect.
So the answer is "no".

---

Table 18: Successful case: our deductive verification approach successfully discovers ungrounded information and reasoning mistakes.

**Input:**
Here is some information:
"How many hours would it take for 5 people to paint a whole house?"

Based on the given information, here is a reasoning process:
"Calculate the total number of hours it would take for 5 people to paint a whole house.
Total hours for one person to paint a whole house: 240 hours
Total number of people: 5
Total hours for 5 people to paint a whole house: 240 hours / 5 = 48 hours"

Double-check the reasoning process, let's analyze its correctness, and end with "yes" or "no".

Answer:
Let's think step by step without any assumptions.

**Output:**
Grounding check
Purpose: Find references for numbers in the reasoning process sequentially
Rule: Any ungrounded number makes the check fails

"240" is a quantity not from the given information, it is ungrounded.
"5" is a quantity from the given information that represents the number of people.
"48" is a quantity calculated from the ungrounded number and grounded number in the reasoning process.
Result: Based on the check, there is one ungrounded number, but it is calculated from another grounded number, so the grounding check passes.

Reasoning check:
Purpose: Check whether the reasoning logic correctly answers the question
Rule: Any logic error makes the check fails

The reasoning process correctly calculates the total number of hours for 5 people to paint a whole house by dividing the total hours for one person to paint a whole house by 5. Therefore, the reasoning check passes.
Calculation check:
Purpose: Check whether the calculation process is consistent
Rule: Any inconsistent calculation makes the check fails

calculation1:
equation: 240 / 5, answer: 48
240 / 5 = 48 is consistent with the answer, so the calculation is correct.
Result: Based on the check, the calculation process is consistent, so the calculation check passes.

Check results: Ground check passes, Reasoning check passes, Calculation check passes.
Rule: All checks pass make the reasoning correct.
So the answer is "yes".

Table 19: Failure case: our deductive verification process fails to find out ungrounded information in the reasoning step. The number 240 in the reasoning step is ungrounded, but the model states that it can be calculated from grounded numbers.

---

**Input:**
Here is some information:
"Calculate the number of doughnuts Derrick makes in a day.
Number of doughnuts in a dozen: 12
Number of doughnuts in 10 dozens: 10 * 12 = 120
Calculate the revenue from selling one doughnut.
Price per doughnut: $2"

Based on the given information, here is a reasoning process:
"Calculate the revenue from selling all the doughnuts in a day.
Number of doughnuts in a day: 120
Revenue from selling all the doughnuts in a day: $2 * 120 = $240"

Double-check the reasoning process, let's analyze its correctness, and end with "yes" or "no".

Answer:
Let's think step by step without any assumptions.

---

**Output:**
Grounding check
Purpose: Find references for numbers in the reasoning process sequentially
Rule: Any ungrounded number makes the check fails

"12" is a quantity from the given information that represents the number of doughnuts in a dozen.
"10" is a quantity not from the given information, it is ungrounded.
"$2" is a quantity from the given information that represents the price per doughnut.
"120" is a quantity not from the given information, it is ungrounded.
Result: Based on the check, there are ungrounded numbers, so the grounding check fails.

Reasoning check:
Purpose: Check whether the reasoning logic correctly answers the question
Rule: Any logic error makes the check fails

The reasoning process is correct. We have the number of doughnuts Derrick makes in a day,
and we have the price per doughnut.
Multiplying these two yields the revenue from selling all the doughnuts in a day.
Result: Based on the check, the reasoning logic is correct.

Calculation check:
Purpose: Check whether the calculation process is consistent
Rule: Any inconsistent calculation makes the check fails

calculation1:
equation: 10 * 12, answer: 120
10 * 12 = 120 is consistent with the answer, so the calculation is correct.
calculation2:
equation: $2 * 120, answer: $240
$2 * 120 = $240 is consistent with the answer, so the calculation is correct.
Result: Based on the check, the calculation process is consistent, so the calculation check passes.

Check results: Ground check fails, Reasoning check passes, Calculation check passes.
Rule: Any failed check makes the reasoning incorrect.
So the answer is "no".

---

Table 20: Failure case: our deductive verification process sometimes treats grounded information as if they were ungrounded. The number 120 is provided in the given information, but the model states that it is ungrounded.

