# OpenReview forum: "Deductive Verification of Chain-of-Thought Reasoning"
_NeurIPS.cc/2023/Conference — NeurIPS 2023 poster_

### Official Review · Reviewer_bmW6 · 2023-07-04

**Soundness:** 3 good
**Presentation:** 4 excellent
**Contribution:** 3 good
**Rating:** 6
**Confidence:** 4

**Summary:**

Despite demonstrating appealing reasoning capabilities, Language Models (LLMs) still suffer from erroneous and hallucinated reasoning. In order to enhance the trustworthiness of LLMs' reasoning process, this paper introduces a deductive reasoning framework based on natural language. The framework extracts premises from the context, breaks down the reasoning process into smaller steps (similar to Chain-of-Thought prompting), and explicitly aims to deduce and justify each step using the extracted premises. The framework is based on prompt engineering and does not use external systems, so it can be categorized into self-verification. This work is mainly benchmarked against previous CoT prompting and faithful CoT.

**Strengths:**

Lack of robustness/trustworthiness is surely one of the main drawback in LLMs' reasoning. The paper presents a plausible framework that resemble the idea in classic symbolic theorem proving and incorporate explainability into LLMs' reasoning process. The paper is nicely structured and with well-crafted examples to illustrate the main idea of this paper at a high level.

**Weaknesses:**

Though the idea of Natural Program is plausible, it would be nice to see slightly more experiments and case analysis, which could further demonstrate the efficacy of this approach. For example, the cited Faithful-COT paper has several other benchmarks like CLUTRR and StrategyQA, and it has also experimented on GPT-4, which showed an amazing advantage over ChatGPT over the benchmarked tasks. It would be interesting to see how well Natural Program performs on advanced models like GPT4.

Another line of related work is Selection-Inference [1] and Faithful Reasoning [2]. I know the authors have cited the later but I believe more comparison is needed as both these works and Natural Program have a premise selection phase. It would also be interesting to see if Natural Program continue to work well on datasets like ProofWriter.

[1] Creswell, Antonia, Murray Shanahan, and Irina Higgins. "Selection-inference: Exploiting large language models for interpretable logical reasoning." arXiv preprint arXiv:2205.09712 (2022).

[2] Creswell, Antonia, and Murray Shanahan. "Faithful reasoning using large language models." arXiv preprint arXiv:2208.14271 (2022).

**Questions:**

- Table 2, what does the Correct/Wrong mean here? By Correct, do you mean that when supplied with correct answer and sampled reasoning chains, the chance of the ChatGPT model giving a yes after being prompted with 'Do you think the above reasoning process is correct? Let’s think step by step'?
- Table 3, Natural Program appears to be particularly good on the AQuA dataset. Could there be some case analysis here?
- Table 4, how are minimal premises decided here? In practice, it is always hard to figure out the minimal set of premises to derive a certain proposition, so I assume the actual performance of the framework will be between Full Premises and Minimal Premises, right?

**Limitations:**

I believe the authors have addressed the limitations well.

---

> ### Author Rebuttal · Authors · 2023-08-09
>
> We sincerely thank you for your constructive feedback and your appreciation of our work! We address the comments and questions below.
>
> > Q1: Add experiments over more benchmarks like CLUTRR and StrategyQA.
>
> Thanks for your suggestion. For StrategyQA, we sample 1000 instances from the dataset. We use 7-shot samples for CoT and 1-shot sample for Natural Program. For CLUTRR, we sample 200 instances from the dataset due to limited rebuttal time. We use 8-shot samples for CoT and 2-shot samples for Natural Program. Both datasets are shuffled with random seed 42.
>
> **Final answer accuracy on StrategyQA and CLUTRR**
> |    Methods   |  StrategyQA | CLUTRR |
> | --- | --- | --- |
> | CoT (few-shot) | 72.45% | 39.25% |
> | Natural Program (1-2 shot) | 69.95% | 46.00% |
>
> > Q2: Add experiments with advanced models like GPT-4
>
> Thanks for your suggestion! It would be a great idea to explore more advanced models like GPT-4. However, due to the high costs associated with GPT-4, particularly when combined with majority voting, we'd like to postpone this exploration to future work when GPT-4 becomes more cost-effective.
>
> > Q3: Related work and comparison with "Selection-Inference" family
>
> Thanks for bringing attention these works and we'll cite "Selection-Inference". We'd also like to note that while both Selection-Inference and Natural Program have premise selection phases, their mechanisms are different. The selection model within the Selection-Inference framework selects facts exclusively from the given context, effectively preventing fact hallucination. On the other hand, Natural Program is more flexible for e.g., instruction-based open-ended reasoning problems, such as Last Letters, which instructs LLMs to output the concatenation of the last letters of all words in a sequence.
>
> > Q4: Evaluate Natural Program on ProofWriter
>
> Thank you for your suggestion. We conduct a random sampling of 100 examples with a depth of 3 from the development subset of the ProofWriter CWA dataset. Both Natural Program and CoT are evaluated using GPT-3.5-turbo (ChatGPT) with a one-shot example as a prompt. The results can be found in the table below.
>
> **Final answer accuracy on ProofWriter**
> | CoT | Natural Program |
> | --- |  --- |
> |  87.00% | 93.50% |
>
>
> > Q5: What is the meaning of "Correct/Wrong" in Table 2?
>
> Thanks for your question. Yes, your understanding is correct. Results in Table 2 illustrate the percentage of verification validity outputs that align with the answer correctness ("Correct/Wrong") of reasoning chains.
>
> > Q6: Results are perticually good over "AQuA".
>
> Thanks for pointing this out. Upon analysis on AQuA, we found that baseline CoT performed worse than Natural Program mainly because its answer responses were not consistently capitalized (e.g., output can be "A" or "a"), which is further because the few-shot prompt from the original CoT paper uses small letters while query questions use capitalized letters. Our previous answer parser only considered capitalized answer choices as correct. Therefore, we adjusted our answer parser to accommodate different capitalizations in answer choices, which improved the accuracies of baselines from our previous Table 3.
>
> We have updated Table 3 in "Common Response". After the update, our Natural Program performs comparably to the CoT baseline on AQuA.
>
> > Q7: How are minimal premises decided in Table 4?
>
> Yes, your understanding is correct. "Minimal premises" means that when verifying each reasoning step from a Natural Program reasoning chain, we only keep the premises whose numbers are cited by that reasoning step as its context for verification. While we encourage LLMs to cite the minimal set of premises when generating reasoning chains in our Natural Program prompt, LLMs might still cite irrelevant premises during actual problem solving processes, thereby introducing irrelevant premises for verification.

---

> > ### Comment · Reviewer_bmW6 · 2023-08-19
> > **Thanks**
> >
> > Thanks a lot for the clarification!

---

### Official Review · Reviewer_kmTd · 2023-07-05

**Soundness:** 2 fair
**Presentation:** 3 good
**Contribution:** 2 fair
**Rating:** 5
**Confidence:** 3

**Summary:**

This paper propose natural program, which a novel method to make verifiable Chain-of-Thoughts reasoning as well as verify each step. The idea is to first generate reasoning steps in the format of '(# by evidences), caculate...'. And then they prompt another LLM to verify each relatively independent step. The paper also propose a method called 'Integrating Deductive Verification with Unanimity-Plurality Voting' to use verification results to improve generation correctness, though they claim it is not their main aim.

**Strengths:**

1. The idea of doing verification on relatively independent steps is good. The results in table 4 shows its verification performance is better than directly using all previous information.

**Weaknesses:**

1. I happen to have read the arxiv version (https://arxiv.org/pdf/2306.03872.pdf) of the paper. But there is a major mismatch between the arxiv version and the submission. Tab. 4 in the Arxiv version has the results of 'Ours (Natural Program (NP), No Verification)' and 'Ours (NP + Deductive Verification + UPV)', which shows Natural Program is helpful but verification and UPV is not. But the submission removed the row
of 'Ours (NP + Deductive Verification + UPV)' and rename the row of 'Ours (Natural Program (NP), No Verification)' as 'Ours (Deductive Verification + NP + UPV)', which can lead to a conclusion that NP + Deductive Verification + UPV is helpful to accuracies. Could you explain this? By the way, where is Tab. 5.2 in the 'Answer correctness' paragraph.
2. I doubt the usefulness of the method. The method cannot be applied to verify reasoning from other generation models, as it requires a specific form of input. The method cannot be used to improve generation accuracy because of the marginal performance gain.

**Questions:**

See weaknesses, especially point 1. If it's my misunderstanding, I will increase my score.

**Limitations:**

The authors have discussed limitations in Sec. 6, which I think is adequate.

---

> ### Author Rebuttal · Authors · 2023-08-09
>
> We sincerely thank you for your constructive feedback! We address the comments and questions below.
>
> We'd like to first clarify that our primary goal is *NOT* to improve the final answer accuracy of reasoning tasks.
>
> When LLMs leverage CoT to carry out problem solving, hallucinations and accumulated errors can often occur in intermediate reasoning steps, even if the final answers are correct (examples illustrated in Tab. 1 of submission). This phenomenon limits the reliability and trustfulness of LLM generations. As a result, we place significant emphasis on ensuring the **validity of every reasoning step**, *not the correctness of the final answer*, such that LLM reasoning outputs become more reliable.
>
> To this end, we first discover that naively using LLM to verify complex reasoning chains results in very poor verification accuracies (see Table 2 in “Common Response”) because of intertwining reasoning-step dependencies that confound LLMs. We therefore propose our Natural Program-based verification approach, which decomposes the verification of an entire reasoning chain into the verification of individual reasoning steps, and more crucially, the verification process of each reasoning step is based on *only* its necessary contexts and premises. As a result, we have *significantly improved the verification accuracy of reasoning chains* (see Table 4 in “Common Response”), thereby significantly improving the reliability of LLM-generated reasoning steps and achieving our primary goal.
>
> > Clarification of Table 3 in our submission and "Tab. 5.2" typo
>
> Thank you for pointing it out. "Tab. 5.2" is a typo, and it refers to Tab. 3 in our submission. We will revise it in the future.
>
> We have updated Table 3 to address the submission-arxiv mismatch. For details, please refer to "Common Response".
>
> > Does deductive verification improve QA/generation accuracy?
>
> Thank you for raising this point. As shown in Table 3, we acknowledge a slight decrease in QA/generation (final answer) accuracy after applying deductive verification on Natural Program reasoning chains. One major contributing factor to this decrease is the filtering out of reasoning chain candidates that provide correct answers but exhibit incorrect deductive reasoning.
>
> However, we'd like to note that with our proposed Natural Program and minimal premise-based verification approach, we have *significantly improved the verification accuracy of reasoning chains* (see Table 4 in “Common Response”, and comparison with Table 2), thereby significantly improving the reliability of LLM-generated reasoning steps and achieving our primary goal.
>
> > Can our deductive verification methods be applied to a broader range of cases, such as other models or different formats?
>
> Thank you for your insightful question. Our Natural Program-based verification approach harnesses the power of natural language, making it potentially applicable to a wide range of natural language reasoning tasks along with any LLM with in-context learning capabilities. For abstract reasoning datasets (e.g., Last Letters), our verification method can be applied when the reasoning processes can be articulated in natural language. For datasets centered around shape or object comprehension, new prompts can be devised to translate shapes and objects into symbols and words, enabling natural language reasoning and verification at the symbol level. For code-based datasets, if natural language reasonings used for code generation are accessible, our approach can be used to verify these reasonings as well.
>
> Furthermore, our verification approach can be applied to fields that naturally demand verification of sequences, such as assessing human reasoning or analyzing arguments in law, which would be interesting for future work.

---

> ### Comment · Reviewer_kmTd · 2023-08-10
>
> Thank you for your reply! I think weakness 1 is mainly solved by your correction and I think vrification is a very important topic. So **I'd love to increase my overall score to 5**.

---

> > ### Author Response · Authors · 2023-08-13
> >
> > We are glad to know that our rebuttal has sufficiently addressed your concerns, and we also truly appreciate your timely response and positive feedback!
> >
> > We would also like to gently remind you that the "review editing" feature has been fixed since your comment was initially submitted (on the first several hours of the author-reviewer discussion period, there was an issue on OpenReview that prevented reviewers from editing their reviews, and this issue has later been resolved). Your consideration in updating the score would be greatly valued!

---

### Official Review · Reviewer_TPbw · 2023-07-07

**Soundness:** 3 good
**Presentation:** 3 good
**Contribution:** 2 fair
**Rating:** 3
**Confidence:** 4

**Summary:**

This paper introduces a format for chain-of-thought reasoning in language models called Natural Programs. In a Natural Program, the model explicitly breaks down the given problem into a numbered list of assumptions. Then, each reasoning step explicitly lists out the premises (starting assumptions or previous steps) that are required to support the conclusion made at that step. The advantage of Natural Programs is that one can verify the reasoning trace by locally verifying each step, extracting the premises and conclusion and asking a language model to assess the validity of that step. A first experiment shows that GPT-3.5 Turbo is not able to do zero-shot verification of full arguments. However, it is much better at catching errors when presented just the minimal set of premises at each step. This allows one to sample multiple Natural Programs and use step-wise verification to select the best among them. The authors experiment on 6 datasets - 4 for mathematical reasoning (GSM8k, AquA, MATH, AddSub) and 2 of commonsense/symbolic reasoning (date manipulation and last latter concatenation), using GPT-3.5 Turbo, showing improvements in final answer accuracy compared to vanilla chain-of-thought with voting and Faithful chain-of-thought.

**Strengths:**

The paper is clear and easy to follow. The idea is well-motivated and timely: chain-of-thought is the dominant paradigm for using LLMs in a variety of tasks, but by itself it's of course not perfect. Also, the paper builds nicely on top of existing LLMs, so ideas like Natural Programs that enhance trustworthiness of pre-trained LLMs have potential for wider adoption.

The idea also builds nicely on the flexibility of LLMs, and this is reflected on the range of tasks used in the paper. Many more domains where CoT is applicable are also amenable to Natural Programs without more effort other than writing adequate prompts for the new domains.

**Weaknesses:**

The paper's first stated contribution is the framework of Natural Programs. However, a similar idea has been proposed in the EMNLP 2022 paper that introduced NLProofS [1]. The idea of NLProofS was also to generate reasoning steps in natural language while having the model explicitly refer to the premises that each step follows from, and then use a verifier to score the validity of each individual step given only the relevant premises. See Figure 1, "Single-shot generation" in [1], comparing that to Figure 1 in the Natural Programs paper. While overall there are differences between both works (e.g., NLProofs assumed a hypothesis to be proved or disproved given at the beginning, which is not given in any of the tasks used in the Natural Programs paper), but I believe this undermines the novelty aspect of Natural Programs.

The authors argue (e.g. line 62, line 253) that their focus is on generating more trustworthy reasoning, rather than just reasoning where the final answer is correct. However, based on the results of Table 4, it does not seem like Natural Programs alone can provide a high level of reliability for GPT-3.5 Turbo. In GSM8k, the best dataset, there's still a 16% false positive rate, then 38% in AQuA, and 60% in AddSub. Thus, it doesn't feel clear to me that this goal has been achieved. I'd thus still place Natural Programs as a means to the end of achieving better final accuracy.

In terms of final-answer accuracy, the results are arguably mixed. The gains over CoT/Faithful CoT are quite significant in AQuA and Last Letters, but minor in the other 4 datasets.

The experiments focus only on a single model, GPT-3.5 Turbo. It would strengthen the paper to show that multiple models can benefit from using Natural Programs. The Appendix does have initial results with Vicuna, but the authors seem to only have done the experiment on validation accuracy, not on final answer accuracy. Even if the absolute performance of Vicuna is lower, perhaps there's a higher chance of obtaining larger gains on it.

[1] Yang, Kaiyu, Jia Deng, and Danqi Chen. "Generating Natural Language Proofs with Verifier-Guided Search." Proceedings of the 2022 Conference on Empirical Methods in Natural Language Processing. 2022.

**Questions:**

- Line 68: It is implied that GPT-3.5 Turbo has 175B parameters, but as far as I'm aware the size of the model is not publicly known (GPT-3 has been reported by OpenAI to have 175B parameters, but not GPT-3.5 Turbo).
- Lines 59-61: What empirical results in the paper support the mentions to 'nearly all' in these sentences? It would be useful to refer to those numbers here. As I understand, there are still a significant number of false negatives when verifying Natural Programs.
- Footnote in Page 7: world problems -> word problems
- What do the authors think are the most significant differences between Natural Programs and the NLProofS approach?
- The Faithful CoT paper reports slightly better numbers for GSM8k (~79% with self-consistency). Are there any differences in the evaluation setup that justify this?
- On average, how many more calls to the model does Natural Programs take, compared to vanilla CoT? This should be mentioned in the paper, since any reader that is a potential user would have this practical question.
- Does verification necessarily require one call per step, or can multiple steps be batched in a single call to the model? It would be interesting if batching with the Natural Programs format still performs better than asking GPT-3.5 to verify the whole solution all at once.
- Is it possible to combine Faithful CoT with Natural Programs in the arithmetic tasks? Perhaps that would allow the model to not need to perform arithmetic, and the verification can also remain more high-level (verifying the reasoning rather than the calculations)
- In line 66, you mention that when all verification fails, the proposed approach can produce "Unknown" rather than a likely wrong answer. How often did this happen in the experiments?

**Limitations:**

Yes, the limitations are clearly discussed in the paper.

---

> ### Author Rebuttal · Authors · 2023-08-10
>
> We sincerely thank you for your constructive feedback! We address the comments and questions below.
>
> > Differences between NLProofS and our work
>
> Thanks for your insightful question and we'll cite the "NLProofS" paper. We'd like to highlight several perspectives that distinguish our work from NLProofS:
> - **The central approach of the two papers differ.** Our approach leverages the in-context learning and step-by-step reasoning abilities of LLMs to perform the verification of each step in a reasoning chain, and does not require LLM finetuning. On the other hand, NLProofS finetunes LLMs on specific datasets to score proof tree nodes and search for proof. Such scoring process does not utilize in-context learning.
> - **The focus of these two papers differ.** The primary goal of NLProofS is to propose "a novel method for stepwise proof generation", while our primary goal is to propose an effective framework to *verify* deductive reasoning chains. To this end, we explore different designs for our verification framework, ultimately leading us to develop the Natural Program reasoning format and adopt the minimal premise-based verification approach.
> - While NLProofS employs a scoring model to assign a validity score to proof tree nodes, it does not provide explainations for **why** a proof step is invalid. In contrast, our Natural Program-based LLM verification approach not only identifies invalid reasoning steps, but they can also provide explicit explanations for why they are invalid, detailing the specific reasoning errors involved.
>
> > Have we achieved the goal of generating more trustworthy reasoning?
>
> Throughout this work, **we have uncovered valuable insights and discovered a verification framework that *significantly* improves the verification accuracy of reasoning chains over naive approaches, and we believe we have achieved our primary objective of improving LLM reasoning reliability**. We are also NOT asserting that the deductive verification problem is completely solved, but we believe that our proposed framework is a promising avenue to explore.
>
> The insights from our paper are as follows: naively using LLM to verify complex reasoning chains results in very poor verification accuracies (see Table 2 in “Common Response”) because of intertwining reasoning-step dependencies that confound LLMs. However, when we break down complex and long logic chains into short ones (e.g., one-step reasoning), and more crucially, verify each reasoning step with *only* its minimal context and premises necessary, we can *significantly* improve verification accuracies (see Table 4), thereby enhancing the reliability of LLM-generated reasonings.
>
> Moreover, we experimentally demonstrate the potential to adapt our insights and our verification frameworks to construct comprehensive verification datasets and further improve LLM verification accuracy through finetuning. In Appendix Table 3, we finetuned Vicuna using our Natural Program verification framework, achieving better verification accuracies. Our Natural Program format and minimal-premise approach may inspire future verification datasets, improving the reliability of LLM-generated outputs.
>
> > I'd still place Natural Programs as a means to achieving better final accuracy. Final answer accuracy results are mixed
>
> While the final answer accuracy results are mixed, we find that **one major contributing factor is the filtering out of reasoning chain candidates that provide correct answers but exhibit incorrect deductive reasoning.** An example is illustrated in Table 6 of submission, where ChatGPT generates the correct final answer but assigns incorrect premise numbers to support the first reasoning step, and our verification approach effectively identifies many of these reasoning errors. Combined with the fact that our proposed verification approach has significantly improved the verification accuracy over naive approaches, we believe we have achieved our primary goal.
>
> > Final answer accuracy on Vicuna
>
> Thanks for your suggestion. We have added final answer accuracy comparision bewteen "Natural Program" and "CoT" for Vicuna models to complement the verification accuracies in Appendix Tab. 3.
>
> |Models|Methods|GSM8K|AQuA|MATH|AddSub|Date|Last Letters|
> |---|---|---|---|---|---|---|---|
> |Vicuna-7B|CoT|9.26%|25.03%|7.25%|28.33%|21.70%|0.02%|
> ||Natural Program|7.79%|23.93%|6.55%|38.10%|27.76%|0.00%|
> |Vicuna-13B|CoT|16.54%|25.26%|7.30%|36.15%|19.79%|1.04%|
> ||Natural Program|34.16%|21.01%|7.09%|31.48%|33.88%|2.36%|
>
> > Number of parameters for GPT-3.5 Turbo
>
> Thanks for your suggestion. We will remove this number.
>
> > L59-61 statement
>
> Thanks for your attention. We'll revise it by replacing "nearly all" with "many".
>
> > Faithful CoT GSM8K results
>
> We rerun Faithful CoT with GPT-3.5-turbo using their original repo and have updated Table 3. It achieves a better performance of 81.20% on GSM8K.
>
> > Number of API calls per question for Natural Program and CoT
>
> For reasoning chain generation, both Natural Program and CoT use 1 API call per question. This is because "Natural Program" and "CoT" both append few-shot examples before the query question.
>
> > Call per step in verification
>
> For each reasoning step, we need one call for verification. We can also aggregate multiple queries into one single call by asking model to answer multiple questions.
>
> > Combine Faithful CoT with Natural Programs
>
> Thanks for your question. Table 3 shows that Faithful CoT does not perform as well as the CoT baseline or our Natural Program across most tasks, likely because Faithful CoT focuses on code generation, which doesn't provide much benefits for our current arithmatic-focused tasks. Due to limited rebuttal time, we'd like to leave further explorations for future work.
>
> > Frequency of "Unknown" (L66)
>
> The frequency of "unknown" is pretty rare (GSM8K: 2.7%; AQuA: 5.0%; AddSub: 2.9%; others: 0%).
>
> > Typo in page 7
>
> Thanks for pointing out! We'll fix it.

---

> > ### Comment · Reviewer_TPbw · 2023-08-14
> >
> > I thank the authors for the several clarifications. I think the new results (e.g. with Vicuna) and incorporating clarifications in the other reviews will improve the paper.
> >
> > I'd like to further clarify two points.
> >
> > First, with respect to NLProofS, I do see all of the differences that the authors wrote about both works (I had noticed some of them in the review, too). All I'm arguing is that the format of Natural Programs, which is claimed as a contribution in the paper, might have to be seen in perspective of related prior work. The abstract of the paper reads: "To facilitate this procedure, we propose Natural Program, a natural language-based deductive reasoning format.". But wouldn't the authors agree that the format itself is essentially what is shown in Figure 1 in the NLProofS paper? I know that the uses are different - NLProofS fine-tuned, this paper focuses on in-context learning, provides explanations while NLProofS only had scores, etc. But if the claim is that the format itself is novel for enabling step-by-step verification, I still think that might need to be rephrased. I'm open to discussing this further, but please note that my remark is narrowly about the Contribution #1 from this paper (L70), not about general differences between both works.
> >
> > Finally, I'm still a bit puzzled by the main result of the paper, as I explain below. I'd like the authors to clarify this point, since otherwise I'm not sure if I'm making sense of the results.
> >
> > In the response, the authors note that the method "significantly improves the verification accuracy of reasoning chains over naive approaches". However, it's also noted that "Unknowns" are quite rare overall.
> >
> > Consider the following scenario, which would be consistent with Table 3: CoT and Deductive Verification achieve similar overall accuracies, but CoT always gives some response whereas Deductive Verification has a much higher calibration - it either gives correct responses, or it fails to verify any chain and gives "Unknown". If this was the case, even if final-answer accuracy wasn't improved, I'd easily see that the authors significantly improved reasoning verification, and this would be a valuable result.
> >
> > But the authors report that Unknowns are rare. This necessarily implies that most of the errors are not "Unknowns", but rather of wrong reasoning chains that nonetheless passed deductive verification. If this is the case, I'm still not seeing how the takeaway can be that the method "has achieved our primary objective of improving LLM reasoning reliability". When models give a wrong answer (which happens at similar rates overall with and without NP + DV), this would imply that both vanilla CoT and the method proposed here still produce an invalid reasoning chain. This does not seem cohesive with the intended takeaway from the human evaluation (Table 4).
> >
> > I'd really appreciate if the authors help me make sense of how these results fit together to support the main claim of the paper.

---

> > > ### Author Response · Authors · 2023-08-15
> > >
> > > Thanks for your reply! We address your follow-up questions below:
> > >
> > > > Comparing the formats of Natural Program vs NLProofS, besides the two work's difference in central goal and central approach mentioned in our rebuttal.
> > >
> > > We’d like to highlight several perspectives that distinguish the format of Natural Program from the format of NLProofS:
> > > - In NLProofS, the initial premises (referred to as "supporting facts" in Fig. 1 of their paper) used for proof search are not extracted through in-context learning by LLMs. Rather, these premises are provided from the datasets NLProofS utilize. On the other hand, in Natural Program, initial premises ("Question-Related Premises" in Fig. 1 of submission) are automatically extracted by LLMs through in-context learning, which makes Natural Program more flexible towards diverse natural language reasoning tasks.
> > > - When deriving a new reasoning step from previous premises, Natural Program is also more flexible, as it allows the use of commonsense knowledge not explicitly listed in premises. For example, consider this problem: "Marin eats 4 apples a day. How many apples does he eat in November?" Even though "November has 30 days" is not explicitly listed in the premises, Natural Program permits the use of such common knowledge within a reasoning step. Our in-context verification process is also capable of handling these implicit premises (e.g., if LLM outputs "November has 29 days" in a reasoning step, it will be marked as invalid).
> > > - NLProofS is limited to tasks that have an explicit proof structure, while Natural Program is compatible with in-context abstract reasoning tasks such as Last Letters, where the LLM is instructed to output the concatenation of the last letters of all words in a sequence as the final answer.
> > >
> > > > How our paper's results fit together to support the main claim of the paper.
> > >
> > > If we are understanding your question correctly, we believe that your question can be rephrased as follows:`Essentially, why does our verification approach significantly improve the verification accuracy of reasoning chains in Table 4 of submission, but barely improves the final answer accuracy in Table 3 of submission?` We answer this rephrased question below:
> > >
> > > Consider the GSM8K dataset as an example (recall that the final answer for a problem is obtained through majority voting). Among all problems, 91.6% of problems have `|number of votes received by the correct answer - largest number of votes received by a single wrong answer| > 2`, and their final answers are unlikely to be changed through our deductive verification approach. For the *rest of the cases (8.4%)*, where deductive verification is more likely to impact their final answers, we found that:
> > > - Among all reasoning chains that arrive at correct answers (these correct-answer chains account for 49.4% of all reasoning chain candidates), 46.2% of reasoning chains are filtered out by our verification process.
> > > - Among the reasoning chains that arrive at correct answer but are filtered out by our verification process, 76.3% indeed exhibit incorrect reasoning.
> > > - Among the reasoning chains that arrive at correct answer and are not filtered out by our verification process, 78.0% indeed have correct reasonings.
> > > - Among the reasoning chains that do not arrive at correct answer and exhibit incorrect reasonings (these account for 50.6% of all reasoning chain candidates), 40.6% are filtered out by our verification process.
> > >
> > > The above statistics shows that a significant portion of reasoning chains that arrive at correct answers but exhibit incorrect reasoning are successfully eliminated. Therefore, the reliability and trustfulness of reasoning chains that arrive at the correct answers are significantly improved. Combined with the fact that a significant proportion of reasoning chains that exhibit incorrect answers are eliminated, and that our approach's verification accuracy significantly improves over naive verification approaches, our primary goal to improve LLM reasoning reliability is accomplished.
> > >
> > > Nevertheless, the removals of many reasoning chains yielding correct answers (specifically, a significant 46.2% x 49.4% of all chains) has a notable impact. This even exceeds the removals of reasoning chains with incorrect reasonings and answers (40.6% x 50.6% of all chains). As a result, there are fewer votes for the correct answer when generating final answers through majority voting, which limits the final answer accuracy. In the future, we believe that when a greater proportion of incorrect reasoning chains with incorrect answers are filtered out, we can improve the final answer accuracy.
> > >
> > > We'll make the above statements more clear in our final draft.

---

### Official Review · Reviewer_BLD6 · 2023-07-08

**Soundness:** 3 good
**Presentation:** 3 good
**Contribution:** 3 good
**Rating:** 6
**Confidence:** 4

**Summary:**

The authors propose Natural Program, a suite of prompt-based steps to verify the reasoning process generated by LLM by reusing the LLM itself. The key observation used is that LLMs are good verifiers natively if they are presented with just the necessary set of premises to judge the statement. Any distractor can potentially increase error in the verification process. Thus, they show the application of their approach on CoT rationales for mathematical datasets. Overall, the approach leads to more trustful explanations and improved accuracy on the tasks.

**Strengths:**

* The observation that using only the least set of premises helps LLMs in self-verification is a good observation. Although, this is not a new finding per say, but the authors present it in a new application scenario, which is interesting.
* The motivation of the problem is well-formulated in the introduction.

**Weaknesses:**

* While the solution works indeed, it is somewhat circular in nature. Since the main hypothesis here is that we don't trust the generations sampled from LLM, then using the same LLM in pretty much similar manner for verification (basic prompting, without any fine-tuning to probably teach the task better) is not very convincing. The solution also relies on sampling multiple outputs for verification and then some sort of voting (see Q2 for more details).
* The utility of the intermediate steps involved in the verification system is not clear. The qualitative analysis is not sufficient to understand why these specific design choices were made or how it helped. Please expand on these experiments to show some ablation of the design choices. You can explore slightly different prompt formats where we don't mention the premise numbers (#6 , #9, etc.), or other such ablations.
* Lines 240-241: There is no justification of this weird design choice of favoring "yes" labels. Why would you have to do this at all? Why not just discard these sentences that lead to ambiguity? How frequent is this scenario? What happens if you start favoring "no" instead? Does it mean the LLMs are not very sensitive to discriminate between yes and no labels?
* Overall, designing new prompts for a task (verification of rationales in this case) seems limited in terms of novelty. This work can perhaps be more suitable for some more NLP-focused conference.

**Questions:**

* Lines 111-112: This statement is too generic. Please note the type of datasets for which it holds. And also back it by pointing to relevant analysis sections or doing some more rigorous analysis.
* Lines 205-207: This part is not clear. Do you sample yes/no values multiple times with the prompt in lines 1186-187 from the verifier and then do some majority voting? Table 5 suggests that you sample multiple times. If so, does it also show that the LLMs are inherently not good at this task and you are essentially relying on some crude statistical hope that the model will get it right most of the time? Isn't that the whole issue being addressed in the paper?

**Limitations:**

* A major component of this approach is the increased overhead in terms of cost of API calls. There is no substantial discussion on how cost efficient / harmful this approach is. Please expand upon this in a separate limitations section.

---

> ### Author Rebuttal · Authors · 2023-08-09
>
> We sincerely thank you for your constructive feedback! We address the comments and questions below.
>
> > Circular nature of using LLM for verification
>
> We'd like to clarify that "the main hypothesis here is that we don't trust the generations sampled from LLM" is actually NOT our hypothesis. Not only do we seek when the LLM cannot be trusted, but also **when it can be trusted**. According to our experiments, when LLM is used as a verifier, although it is not trustable if the logic chain to verify is complex and long, it is, indeed, highly trustable for short logic chains (e.g., one-step reasoning). This is the motivation for us to invent the Natural Program which allows us to break complex and long logic chains to short ones so that we can trust the verification result of LLMs. In fact, this well aligns with how humans verify results -- recall the experience when we were elementary school students, teachers ask us to write down reasoning steps in very details so that the logical flaws of the answers can be easily detected.
>
> These insights are validated by Table 2 (see "Common Response"), which shows poor verification accuracies when naively using the LLM to verify reasoning with entangled premises. Conversely, Table 4 (see "Common Response") illustrates how our Natural Program-based approach with decomposition and simplification leads to improved verification accuracy and fewer reasoning errors.
>
> We also share the view that constructing comprehensive verification datasets and finetuning LLMs on these datasets can further enhance their verification accuracy. In Appendix Table 3, we finetuned Vicuna using our Natural Program verification framework, achieving better verification accuracies. Our Natural Program format and minimal-premise approach may inspire future verification datasets, improving the reliability of LLM-generated outputs.
>
> > Intermediate steps and premise numbers for verification: ablations and design choices
>
> Premise numbers serve to encourage LLMs to cite the *minimal* set of premises necessary for each generated reasoning step as they work through problem-solving. In this way, when LLMs verify each reasoning step, only the cited premises are utilized as the context for verification. Many irrelevant premises, which can easily confound and distract LLMs, are removed. This process is illustrated in Figure 1 of the paper. Without premise numbers, it is very challenging to remove irrelevant premises when verifying each reasoning step.
>
> In Table 2 (refer to "Common Response"), we perform an ablation by excluding premise numbers from the reasoning generation prompt (prompt in Table 10 in the Appendix). Here, despite LLM attempting to verify each reasoning step, irrelevant premises remain during step verification. In contrast to Table 4 (minimal premises - our approach), verification accuracy experiences a notable decline.
>
> Another ablation is in Table 4, where we keep the premise numbers in the reasoning chain generation prompt. However, during step verification, premise numbers are disregarded, and all preceding steps are used as verification context (e.g., verifying step "#6" includes all prior steps "#1...#5" as context). This is seen in the "Full Premises" entry, where verification accuracy markedly decreases.
>
> > L240-241 - choice of using "yes" as the default verification results when the model does not provide a direct answer
>
> Firstly, LLMs generally output "yes" or "no" following our few-shot examples and rarely output other answers  (GSM8K: 0.04%; AQuA: 0.19%; MATH: 0.16%; AddSub: 0.00%; Date: 1.30%; Last Letters: 1.3%). Our voting strategy treats outliers as a separate category, further minimizing their impact on our approach's efficacy.
>
> Additionally, we manually checked the model's reasoning steps when no direct answer is provided, finding them correct in nearly all cases on our dataset, so the verification answer is 'yes'. This is reasonable, as LLMs can answer most problems in the datasets we use.
>
> > Novelty of prompting
>
> Firstly, we'd like to note that two notable papers accepted by NeurIPS 2022 on LLMs primarily focused on prompting for step-by-step reasoning: "Chain-of-Thought Prompting Elicits Reasoning in Large Language Models" with 1010 citations as of Aug 9, and "Large Language Models are Zero-Shot Reasoners" with 485 citations as of Aug 9.
>
> We'd also like to note that our work goes beyond studying prompts for LLM reasoning chain verification. A crucial component of our framework involves simplifying the step-by-step verification process by only keeping necessary premises and contexts for each reasoning step.
>
> > L111-112 too generic
>
> We acknowledge that this statement is too generic. We'll revise it as "We observe that for many cases where LLMs produce erroneous final answers through a step-by-step reasoning process, there exists at least one mistake among the intermediate reasoning steps."
>
> > L205-207 clarification
>
> We'd like to clarify that we are NOT "relying on crude statistical hope that the model will get it right most of the time", or relying on extensive voting.
> - While we can sample multiple yes/no values for verification and perform majority voting, Table 5 (see "Common Response") shows that while multiple votes offer some improvement, the gains are modest compared to using a single vote.
> - Compared to voting, the premises used for verifying of each reasoning step plays a *much* larger role, as shown in Table 4. Removing irrelevant premises that confound LLMs significantly improves verification accuracy.
>
> > Limitations of API calls
>
> We recognize that including the verification step increases API calls compared to omitting it. Though, in situations like legal cases and answering scientific questions, reliability of reasoning is often prioritized over economic factors. We'll add discussion.
>
> Also, Table 5 shows that adding more votes for verification provides slightly more benefits, but does not differ much from using one vote.

---

> > ### Comment · Reviewer_BLD6 · 2023-08-11
> > **Response to Authors**
> >
> > Thanks for addressing my concerns and clarifying my doubts. Given the ablations, I'm convinced that the prompt designs in the paper are meaningfully contributing to the performance. While this idea of "simplifying the step-by-step verification process by only keeping necessary premises and contexts for each reasoning step", i.e., selecting relevant information and then using for reasoning, is explored in prior works [1,2], I believe this is a slightly new area where the authors show the benefit of such approach. Overall, I'm inclined to accept the paper.
> >
> > [1] Selection-Inference: Exploiting Large Language Models for Interpretable Logical Reasoning, Creswell et al., 2022
> >
> > [2] FaiRR: Faithful and Robust Deductive Reasoning over Natural Language, Sanyal, et al., 2022

---

> > > ### Author Response · Authors · 2023-08-13
> > >
> > > We are glad to know that our rebuttal has sufficiently addressed your concerns and doubts, and we also truly appreciate your timely response and positive feedback!

---

### Official Review · Reviewer_LzaJ · 2023-07-10

**Soundness:** 3 good
**Presentation:** 3 good
**Contribution:** 3 good
**Rating:** 6
**Confidence:** 4

**Summary:**

The authors present an approach to build easily verifiable prompted systems by proposing a new reasoning format - Natural Program. In this proposed form, LLMs are expected to generate a rationale but additionally also indicate previous steps that led to each step of reasoning. This allows a verifier to only look at these dependent steps and reduce the amount of distraction. This is an interesting idea and as the authors show results in an improvement in verification accuracy (verifiers are prompted LLMs too).


**Strengths:**

- A natural language formalism that allows us for more accurate reasoning verification
- The approach is well-motivated by showing the inability of ChatGPT to identify errors given the entire reasoning chain.
- Shows gains compared to self-consistency with two different CoT formats


**Weaknesses:**

- The empirical results could be strengthened with strong baselines. E.g. the paper only has two baselines CoT and FaithfulCoT. See questions for more details.


**Questions:**

- Isn't "unanimity-plurity" just self-consistency with a deductive validation filter? Not sure if it needs another name.

- I am surprised that CoT + Voting achieves a much higher score than Faithful CoT + voting. Also the reported number here seems to match the base numbers for faithfulCoT (no voting). See https://github.com/veronica320/Faithful-COT.

- I would have preferred that all the tables included accuracy numbers on the datasets (instead of just verification accuracy).

- Also I didn't see the value of splitting the numbers into accuracy on correct vs wrong chains. Note that I do like the evaluation set, but you could just report the F1 score. Currently it is hard to see which approach is better when looking at two numbers.

- Since FaithfulCoT also provides a similar dependency structure, could deductive verification be applied with their approach too (as a baseline)?

- While the authors show that full premises (i.e. w/ distractors) does result in lower verification accuracy, it is unclear that it impacts QA accuracy. E.g. Could CoT with UPV already outperform their approach without needing NP? Basically sample K CoTs, validate each step of the CoT given the entire context using K' generations and then pick the validated CoTs for majority voting.


**Limitations:**

Natural programs can introduce errors via incorrect dependency structure prediction. It would have been nice to see an error analysis in the paper to capture the source of errors with NP.

---

> ### Author Rebuttal · Authors · 2023-08-07
>
> We sincerely thank you for your constructive feedback! We address the comments and questions below.
>
> > Q1: Include stronger baselines for comparision with "Natural Program".
>
> We'd like to note that our primary goal is to improve the *truthworthiness and reliability* of the reasoning via step-by-step deductive verification. As such, we don't primarily focus on comparing the final answer correctness of "Natural Program" with other established prompting techniques. Our choice of CoT and Faithful CoT as baselines in Table 3 stems from two reasons:
> 1) When we require the LLMs to generate output in the "Natural Program" format, we utilize few-shot prompting. Consequently, we included "CoT", which also employs step-by-step few-shot prompting for reasoning.
> 2) "Faithful CoT" and our proposed "Natural Program" both require LLMs to explicitly generate premises for each reasoning step. Unlike "Natural Program" that produces step-by-step answers in natural language, "Faithful CoT" generates a step-by-step code program as answers and executes the program with interpreters. We empirically find Faithful CoT to underperform Natural Program, suggesting that generating intricate code programs alongside reasoning chains could be challenging, while generating natural language-based reasonings could be comparatively simpler for LLMs.
>
> > Q2: Why "Unanimity-Plurality Voting"?
>
> We named our voting method "Unanimity-Plurality Voting" because there are two phases of majority voting: the "unanimity" phase and the "plurality" phase.
> - The "unanimity" phase of the majority voting occurs when verifying a reasoning chain. During this phase, (1) To verify each reasoning step $s_i$ of a reasoning chain $S$, we sample a set of $k'$ validity predictions, and perform majority voting to obtain the validity prediction for $s_i$; (2) For a reasoning chain $S$ to be valid, all its reasoning steps have to be "unanimously" valid.
> - The "plurality" phase of the majority voting occurs when we produce final answer for problem solving. During this phase, we perform majority final answer voting over the validated reasoning chains from the "unanimity" phase.
>
> > Q3: Faithful CoT + Voting result update; Why does "CoT + Voting" outperform "Faithful CoT + Voting"?
>
> We have updated Table 3 where the numbers in Faithful CoT reflect majority voting results (please see "Common Response"). From the new Table 3, we still find that CoT sometimes outperforms Faithful-CoT. This difference might stem from the nature of the methods: CoT provides natural language answers, whereas Faithful-CoT generates code solutions. For challenging tasks such as MATH and GSM8K, the complexity of correct code programs could be substantial, potentially making natural language solutions simpler for LLMs to generate.
>
> > Q4: Include final answer accuracy to the tables besides verification accuracy.
>
> Thanks for your suggestion! We will position Tables 3 (final answer accuracy) and 4 (verification accuracy) on the same page. We'll also add final answer accuracies to Table 2.
>
> > Q5: Report F1 score of deductive verification
>
> Thanks for your suggestion! We have updated Tables 2, 4, and 5 to include F1 score. Please see "Common Response" for further details.
>
> > Q6: Deductive Verification of "Faithful CoT"
>
> Thanks for the suggestion! We've conducted an experiment on the GSM8K dataset to show the deductive verification performance of "Faithful CoT". In a manner similar to the verification evaluation pipeline for "Natural Program", we have manually chosen 100 reasoning chains—50 deductively valid and 50 with reasoning errors. We validated each reasoning step using full premises and minimal premises. Detailed results are presented in the subsequent table.
>
> **Verification results of GSM8K with Faithful CoT and Natural Program**
> | Answer Correctness| Faithful - Full reasoning chain | Faithful - Step-by-step (Minimal premises) | NP - step-by-step (Minimal preimises)|
> |--|---|---|---|
> |Correct|96.00%|96.00%|84.00%|
> |Incorrect|12.00%|18.00%|84.00%|
> |F1|44.15%|49.28%|84.00%|
>
> > Q7: Using UPV with CoT without Natural Program (NP)
>
> Thanks for the suggestion! Unfortunately, for reasoning chains generated with vanilla CoT, it is very challenging to extract the actual premises of a single reasoning step because of intertwining reasoning-step dependencies. In contrast, Natural Program allows individual reasoning steps and their minimal set of premises to be easily extracted. Therefore, step-by-step deductive verification is more compatible with answers generated through NP than vanilla CoT. As for UPV, it also depends on step-by-step verification results, and thus has better applicability with NP.
>
> > Limitations
>
> Thanks for the suggestion! We'd like to point the reviewer to Sec. 6 and Tab. 7, where we showcase our approach's failures to detect reasoning step errors when encountering ambiguous wordings. We also included more success / failure analysis in Appendix D and Table 12-14 of Appendix.

---

> > ### Comment · Reviewer_LzaJ · 2023-08-21
> > **Thanks**
> >
> > Thank you for the response. Given the additional emphasis on "encouraging the validity of each reasoning step", this work is better compared with other techniques that evaluate the validity of reasoning e.g. ROSCOE[1], RecEval [2]. These methods also don't rely on any special structure and use models to identify reasoning errors. They could also be used with UPV to improve accuracy.
> >
> > > very challenging to extract the actual premises of a single reasoning step because of intertwining reasoning-step dependencies.
> > I agree and hence my suggestion to view every CoT statement depends on all previous statements. I understand that Table 2 evaluates the validation accuracy but I would also like to see its impact on QA accuracy (assuming you still use UPV)
> >
> >
> >
> > [1] ROSCOE: A Suite of Metrics for Scoring Step-by-Step Reasoning (released last year)
> > [2] ReCEval: Evaluating Reasoning Chains via Correctness and Informativeness (its fair to not compare against this work as it is too new)

---

> > > ### Author Response · Authors · 2023-08-21
> > >
> > > Thanks a lot for your reply! We'll cite ROSCOE and ReCEval. We'd also like to highlight several perspectives that distinguish our work from ROSCOE and ReCEval:
> > > - Our Natural Program-based approach leverages the in-context learning and step-by-step reasoning capabilities of off-the-shelf LLMs to extract individual reasoning steps, minimal premises for each reasoning step, and conduct step-wise verification. **Importantly, our approach achieves the entire process without the need for LLM finetuning**. On the other hand, both ROSCOE and ReCEval rely on finetuned language models for verification (ROSCOE relies on finetuned SimCSE embedding model on specific datasets; ReCEval relies on finetuned T5 model on gold reasoning chains of specific datasets)
> > > - While both ROSCOE and ReCEval assign validity scores to reasoning chains, they do not provide explainations for **why** a reasoning step is invalid. In contrast, our Natural Program-based LLM verification approach not only identifies invalid reasoning steps, but they can also provide explicit explanations for why they are invalid, detailing the specific reasoning errors involved.
> > > - Our Natural Program-based reasoning and verification approach is compatible with in-context abstract reasoning tasks where reasoning steps do not possess proof-like entailment structures. For example, our approach is compatible with Last Letters, where the LLM is instructed to output the concatenation of the last letters of all words in a sequence as the final answer.
> > > - Our Natural Program approach allows the use of commonsense knowledge not explicitly listed in premises. For example, consider this problem: "Marin eats 4 apples a day. How many apples does he eat in November?" Even though "November has 30 days" is not explicitly listed in the premises, Natural Program permits the use of such common knowledge within a reasoning step. Our in-context verification process is also capable of handling these implicit premises (e.g., if LLM outputs "November has 29 days" in a reasoning step, it will be marked as invalid).

---

### Author Rebuttal · Authors · 2023-08-09

# Common Response

We sincerely thank all reviewers and ACs for their constructive feedback! We're glad that reviewers found our paper well-motivated (`LzaJ, BLD6, TPbw`), our verification framework and ideas plausible (`bmW6`) and good (`kmTd`), our observations intersting (`BLD6`), and our papers clear and easy to follow (`TPbw, bmW6`).

We address common reviewer comments here and provide a revision of our tables.

> Primary goal of this work

We'd like to first clarify that our primary goal is *NOT* to improve the final answer accuracy of reasoning tasks. Our primary goal is to encourage the **validity of each reasoning step**, emphasizing LLM reasoning reliability over final answers.

To this end, we first discover that naively verifying complex reasoning chains with LLMs yields low accuracy (see Table 2) because of intertwining reasoning-step dependencies that confound LLMs. We thus propose Natural Program-based verification, decomposing chains into steps and verifying each step with *only* its necessary context and premises. This significantly improves verification accuracy (see Table 4), enhancing LLM-generated reasoning's reliability and achieving our primary goal.

> Adding F1 metric to Tables 2, 4, 5

We appreciate Reviewer LzaJ's suggestions. We have added F1 scores to Tables 2, 4, and 5. When calculating F1 scores, for reasoning chains that are correct, a "true positive" is defined as a verification approach outputing "yes" for the final validity answer; for reasoning chains that are incorrect, a "true positive" is defined as a verification approach outputing "no" for the final validity answer.

**Table 2**: Two-shot reasoning chain verification accuracy for GPT-3.5-turbo with a naive verification approach, where we input an entire reasoning chain for verification, and premises of different reasoning steps are entangled together. Prompt in Appendix Table 10.

|Answer Correctness|GSM8K|AQuA|MATH|AddSub|Date|Last Letters|
|-|-|-|-|-|-|-|
|Correct|98.00%|96.00%|100.00%|92.00%|100.00%|96.00%|
|Incorrect|2.00%|4.00%|0.00%|6.00%|26.00%|6.00%|
|F1|35.03%|36.58%|33.33%|37.43%|57.13%|38.56%|

**Table 4**: 1-shot deductive verification accuracy for GPT-3.5-turbo using our Natural Program-based approach with step-by-step reasoning chain decomposition. During the verification of each reasoning step, we consider two options: utilizing the complete premises $p_i=QC \cup S_{\le j}$, or keeping the minimal set of premises $\bar{p_i} \subseteq p_i$.

|Premise Context|Answer Correctness| GSM8K|AQuA|MATH|AddSub|Date|Last Letters|
|-|-|-|-|-|-|-|-|
||Correct|64.00%|54.00%|58.00%|95.00%|26.00%|96.00%|
|Full Premises|Incorrect|56.00%|68.00%|56.00%|24.00%|76.00%|5.00%|
||F1|59.94%|60.81%|56.70%|53.66%|47.73%|37.58%|
||Correct|84.00%|72.00%|70.00%|95.00%|90.00%|96.00%
|Minimal Premises|Incorrect|84.00%|62.00%|76.00%|40.00%|56.00%|6.00%|
||F1|84.00%|66.92%|72.98%|64.84%|72.20%|38.56%|

**Table 5**: Ablation of different values of k′ on the verification accuracy of reasoning chains using our Unanimity-Plurality Voting strategy. Experiments are performed on AddSub using GPT-3.5-turbo.

|Answer Correctness|k′=1|k′=3|k′=5|k′=10|
|---|---|---|---|---|
|Correct|86.00%|90.00%|90.00%|92.00%|
|Incorrect|38.00%|38.00%|38.00%|40.00%|
|F1|59.68%|61.39%|61.39%|63.53%|


> Revision of Table 3

**The major goal of this work is to improve the trustworthiness and reliability of reasoning. Table 3 aims to illustrate that, along this process, our deductive verification approach does not significantly compromise the accuracy of the final answers.**

We update Table 3 results here and will update our submission pdf in the future. For the baselines (CoT and Faithful CoT), we adopt the implementations from their original repositories and report the results. These baselines utilize varying numbers of few-shot samples across datasets, exceeding our approach's usage (we use 2-shot for Date and 1-shot for other datasets). Additionally, for AQuA, we observed an inconsistency in capitalization between the baseline CoT's official few-shot prompt and the AQuA answer choices. This discrepancy led the baseline CoT to generate answers with random capitalizations. Our previous answer parser only considered capitalized answer choices as correct, so we adjusted our answer parser to accommodate different capitalizations, which improved the accuracies of baselines.

We find that prompting language models to reason in our Natural Program format sometimes achieves better final answer accuracy than baselines. Upon further applying our deductive verification approach to filter out invalid reasoning chains, we observe a slight decrease in final answer accuracy. **One major contributing factor to this decrease is the filtering out of reasoning chain candidates that provide correct answers but exhibit incorrect deductive reasoning**. An example is illustrated in Table 6 of our submission, where ChatGPT generates the correct final answer but assigns incorrect premise numbers to support the first reasoning step. We note that in many such cases, our verification approach effectively identifies these reasoning errors, thereby enhancing the rigor and reliability of the language models’ reasoning processes, albeit with a slight negative impact on the overall final answer correctness.

**Table 3**: Final answer accuracy comparison on GPT-3.5-turbo. All approaches generate $k = 10$ reasoning chains for each problem before performing majority voting or reasoning chain filtering with our deductive verification approach.

|Methods| GSM8K|AQuA|MATH|AddSub|Date|Last Letters|
|-------|------|----|----|------|----|------------|
|CoT + Voting, few-shot|87.62%|70.18%|35.93%|92.36%|69.97%|81.60%|
|Faithful CoT + Voting, few-shot|81.20%|61.80%|31.78%|88.35%|73.50%|-|
|Ours (Natural Program (NP), No Verification), 1-2 shot|87.05%|70.34%|36.75%|93.67%|72.49%|92.98%|
|Ours (NP + Deductive Verification + UPV), 1-2 shot|86.01%|69.49%|36.48%|93.54%|71.45%|92.60%|

---

### Decision · Program_Chairs · 2023-09-21

**Decision:**

Accept (poster)

**Comment:**

While the high-level idea of Deductive Verification of CoT motivating this paper is somewhat straightforward, it is far from trivial to work out details that make such a deductive approach work in practice and I (along with many reviewers) are happy to see such contributions along with the experimental results.  Paraphrasing from a reviewer comment: the topic is interesting, the authors test on a variety of interesting domains, and the approach is adaptable.  In the end, the majority of reviewers agreed to recommend acceptance but also recommend the authors to incorporate critical clarifications that came up in the reviews/rebuttal/discussion.

I won't mention all points of clarification, but I do want to highlight two particular key points where clarification on revision will improve the paper:

Reviewer TPbw and BLD6 both raise issues concerning circularity in defining validity in terms of the LLM itself in a zero-shot manner.  As one reviewer states: "the paper could benefit from a better understanding of when and why validity =/=> correctness".  While the reviewer perspective may differ from the author position here, these issues should be acknowledged and discussed on revision along with potential suggestions for future work that may mitigate any concerns.

Further, the final comments in response to reviewer TPbw w.r.t. NLProofS and reasoning chains are very helpful and should be incorporated in the final revised paper.